# SPECULATIVE SAMPLING FOR PARAMETRIC TEMPORAL POINT PROCESSES

## ABSTRACT

Temporal point processes are powerful generative models for event sequences that capture complex dependencies in time-series data. They are commonly specified using autoregressive models that learn the distribution of the next event from the previous events. This makes sampling inherently sequential, limiting efficiency. In this paper, we propose a novel algorithm based on rejection sampling that enables exact sampling of multiple future values from existing TPP models, in parallel, and without requiring any architectural changes or retraining. Besides theoretical guarantees, our method demonstrates empirical speedups on real-world datasets, bridging the gap between expressive modeling and efficient parallel generation for large-scale TPP applications.

## 1 INTRODUCTION

Event data is prevalent in social networks, natural phenomena, and financial transactions. Events occur irregularly which poses unique challenges, particularly as the timing of events is often influenced by the history. For example, aftershocks follow earthquakes and replies follow messages. Event data often has high sampling frequency with thousands of events per second. Since every millisecond counts, it is crucial to have a scalable sampling method while capturing the true process.

Temporal point processes (TPPs) are the canonical framework for modeling events, they generate sequences consisting of event types (marks) and arrival times on some time interval. The most common implementation is an autoregressive model, where the history of events informs the prediction of the next event. Consequently, the sampling process is sequential which can be inefficient, especially in high-frequency data (Aït-Sahalia & Jacod, 2014), leading to bottlenecks in real-time applications.

Neural TPPs are primarily designed as autoregressive models (Shchur et al., 2021), similar to their counterparts in time series and language modeling (Salinas et al., 2020; Radford et al., 2019). Some recent works propose alternatives that learn to predict multiple future steps instead of only one (Gloeckle et al., 2024; Zeng et al., 2023; Lüdke et al., 2023). Our approach takes a different route, we introduce a new sampling procedure that does not require altering or retraining the underlying model. This allows us to improve the efficiency of existing parametric autoregressive models.

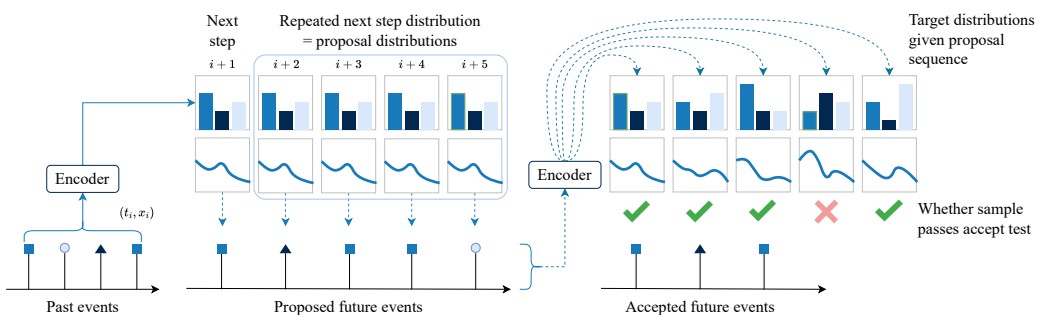

Figure 1: Illustration of our speculative sampling method for temporal point processes.

Figure 1 illustrates our approach. A pretrained encoder predicts the distribution of the next event based on historical data. We reuse this prediction as a proposal distribution to generate multiple future events simultaneously. The same encoder inputs all proposed samples, in parallel, producing the target distributions. We accept proposed samples until we encounter the first event where the proposal and target distributions diverge. In Figure 1, this occurs at the fourth step, accepting the first three proposed events. In the paper we rigorously show this is an exact sampling procedure.

Our main contributions are: **(1)** a novel method for sampling multiple future events that can be seamlessly applied to many TPP models. Our method addresses practical problems in real-world domains like finance, making it relevant to practitioners. **(2)** We propose a way to compute the rejection sampling constant for most common distribution choices and provide a theoretical foundation for our approach. **(3)** We show improvements in sampling efficiency by conducting experiments on widely used benchmark datasets. Additionally, the results provide new insights into these datasets. **(4)** We highlight a particularly suitable application of our technique in the financial domain.

## 2 BACKGROUND

### 2.1 TEMPORAL POINT PROCESSES

Temporal point processes (TPPs) (Daley & Vere-Jones, 2006) are stochastic processes whose realizations are event sequences $\boldsymbol{x} = (x_1, \ldots, x_n)$, $x_i \in \{1, \ldots, D\}$ observed at strictly increasing arrival times $\boldsymbol{t} = (t_1, \ldots, t_n)$, $0 < t_1 < \cdots < t_n \le T$. That is, each event is a random point in time $t_i$ with an assigned event type $x_i$, called a mark. The future events often depend on the past, so we denote the history of the $i$th event as $\mathcal{H}_i = \{(t_j, x_j) : t_j < t_i\}$, a set of all the events that came before $t_i$.

One way to specify a TPP is with an intensity function $\lambda(t)$ which tells us about the concentration of points around each time location $t$. A trivial example is the constant intensity which gives rise to a homogeneous Poisson process. A more expressive *conditional* intensity function incorporates the information of the past events. A completely equivalent parameterization is defining the density function $p(\tau)$, on inter-event (delta) times $\tau_i = t_i - t_{i-1}$, with $t_0 = 0$ (Shchur et al., 2020).

When using a density parametrization, the existing machinery makes it very easy to specify the likelihood and the sampling is straightforward. In the following, we use density-based framework in order to define speculative sampling approach for TPPs. On the other hand, non-parametric intensity models require Monte Carlo in training and they use thinning algorithm for sampling, making them less applicable for our approach.

For practitioners, parametric TPPs are often the default choice due to ease of use without making compromises on performance. Our proposed method adheres to this same principle: it is applicable to any existing parametric TPP model without requiring retraining or model modifications, while maintaining exact sampling and accelerating the sampling process.

A parametric TPP model is specified as $p(\tau_i, x_i | \mathcal{H}_i)$, joint distribution over the next ($i$th) time point and its mark, conditioned on the history. Neural TPP models usually encode the history with a neural network that returns a fixed-sized vector representation $\boldsymbol{h}_i \in \mathbb{R}^h$. Examples of encoders include recurrent neural networks (RNNs) (Cho et al., 2014) and transformers (Vaswani et al., 2017). Our proposed sampling method is model agnostic and we use established models for our experiments.

The model is trained by maximizing the log-likelihood (Daley & Vere-Jones, 2006):

$$\log p(\boldsymbol{t}, \boldsymbol{x}) = \sum_{i=1}^{n} \log p(\tau_i, x_i | \mathcal{H}_i) + \log S(\tau_{n+1} | \mathcal{H}_n), \quad \tau_{n+1} = T - \tau_n, \tag{1}$$

where $S(\tau_{n+1})$ denotes the survival function, the probability that no event occurred since the last ($n$th) event and until the end of the observed interval $T$. The likelihood of the whole dataset is a product of all individual sequence likelihoods.

Conventional sampling from the model starts with the existing events $\{(t_1, x_1), \ldots, (t_n, x_n)\}$. This sequence is processed with a neural network to obtain the distribution of the next delta time $p(\tau_{n+1})$ and mark $p(x_{n+1})$. One can sample $\tau_{n+1}$ and $x_{n+1}$ directly from these distributions, append them to the existing sequence, and repeat this process until some stopping criterion is met.

## 2.2 REJECTION SAMPLING

Sampling from a density function is straightforward. Traditionally, TPPs specified with an intensity function use a different approach called thinning Ogata (1981). Given an intensity function $\lambda(t)$ for which we know the upper bound $\lambda(t) < \lambda_{\max}, \forall t$, we can sample points using the following steps:

1. Sample candidate points $t_i$ from a proposal homogeneous TPP with intensity $\lambda_{\max}$,
2. Compute $\lambda(t_i)$ under the target process and draw a random value $u_i \sim (0, \lambda_{\max})$,
3. Keep the point $t_i$ if $u_i < \lambda(t_i)$, else remove the point (thin).

This approach works because the *proposal* process intensity is larger than the *target* process intensity on the whole domain. Then each sample is kept with the probability proportional to the intensity of the target process. This is an example of rejection sampling applied to TPPs.

Rejection sampling is a general method of obtaining samples from a target distribution by accepting and rejecting samples from some proposal distribution; see, e.g., Devroye (1986) or Bishop & Nasrabadi (2006, Chapter 11) for an overview. Given a proposal distribution $g(x)$ and a target distribution $f(x)$, a sample $x \sim g(x)$ is accepted as a sample from $f(x)$ with probability $f(x)/(Mg(x))$, where $M$ is the upper bound on the ratio $f/g$. Note that $f$ and $g$ do not have to be normalized, but $g$ does have to dominate $f$, which is why we incorporate $M$ to ensure that the ratio is always lower than 1. In case $f = g$, the probability of acceptance will be 1 since $M = 1$. If a proposal distribution is close to the target, we get low rejection rates and efficient sampling.

To summarize, the prerequisite for rejection sampling, given a target density $f(x)$ and a proposal density $g(x)$, is that we can evaluate both $f(x)$ and $g(x)$, we can readily sample from $g(x)$, and that we know $M = \max_x f(x)/g(x)$. Since most common distributions allow density evaluation and sampling, our focus is on finding the rejection constant.

## 3 METHOD

### 3.1 REJECTION CONSTANT FOR SELECTED DISTRIBUTIONS

The **categorical distribution** is one example of a distribution where we can evaluate the rejection constant $M$ directly. The distribution is defined on the space of $D$ categories, where each category $x$ has a probability $p(x)$ of occurring. For a target and a proposal distribution defined with $p_T$ and $p_P$, respectively, we obtain $M$ by evaluating the ratio for all $D$ possible values and take the maximum:

$$M = \max_{x \in \mathcal{X}} \frac{p_T(x)}{p_P(x)}, \quad \mathcal{X} = \{1, 2, \ldots, D\}. \tag{2}$$

Note that some low probability categories can dramatically influence $M$. For example, having a two-class proposal distribution with probabilities $\boldsymbol{p}_P = [1 - \epsilon, \epsilon]$; and a target distribution with probabilities $\boldsymbol{p}_T = [1 - c\epsilon, c\epsilon]$, we get rejection constant of $c$, even for very small $\epsilon$. We propose excluding categories with the highest ratios when they contain less than $\delta$ of the target distribution's probability mass. If $\delta$ is small, the resulting samples closely follow the target distribution. In fact, we guarantee an upper bound on the total variation distance between the approximated and true distributions to be equal $\delta$. Appendix A.3 provides a more rigorous treatment with an implementation.

This means we can choose to use truly exact sampling or introduce a controllable error into the sampling procedure for better performance.

For an **exponential distribution** whose probability density function (PDF) is given by $f(x; \lambda) = \lambda e^{-\lambda x}$, $\lambda > 0$, we can write the ratio of two exponential density functions as:

$$\frac{f_T(x)}{f_P(x)} = \frac{\lambda_T}{\lambda_P} e^{(\lambda_P - \lambda_T)x}.$$

This is a monotonically decreasing function when $\lambda_P < \lambda_T$, giving us the maximum ratio at $x^\star = 0$. However, when $\lambda_P > \lambda_T$, the ratio is unbounded and we cannot evaluate it for all the points on the domain. One way to solve this is to restrict the domain similar to the categorical distribution. For example, we can provide coverage up to 99th percentile of $f_T(x)$, evaluating the ratio at this point.

Not all distributions permit such straightforward analysis, examples include most distribution mixtures. Because of this we devise an alternative approach, described in the following.

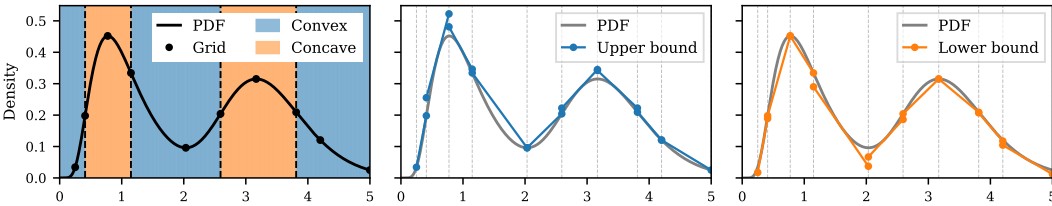

Figure 2: Illustration of upper and lower linear bounds of a mixture distribution with two components. The grid points always include inflections which makes the rejection sampling exact.

## 3.2 General rejection constant

The main idea is to upper bound the target density function and lower bound the proposal. Since we are free to choose bounding functions, we will choose those that allow us straightforward evaluation of the ratio between them, which consequently gives us a way to find the rejection constant (Theorem 3.1).

We decide to approximate a density function with a piecewise linear function, which will allow us simple computation of the rejection constant. This is similar to an envelope construction which has been previously studied for log-concave distributions (Gilks & Wild, 1992). The difference to previous works is that our general method works with many choices of densities, including mixtures of distributions, and is efficient to compute on modern hardware. See Section 4 for further discussion.

We construct linear segments on a grid $\{x_0, x_1, \ldots, x_n\}$. For example, exponential density is convex so it can be upper bounded by the line segment connecting $(x_i, f(x_i))$ and $(x_{i+1}, f(x_{i+1}))$. Alternatively, it can be lower-bounded by a tangent $f'(x_m)$ passing through a midpoint $x_m = \frac{x_i + x_{i+1}}{2}$, for all segments. If the function is concave, these bounds are reversed. From this we can bound any well-behaved density by simply decomposing its domain into convex and concave regions (Lemma A.2). A valid grid needs to, at a minimum, contain all the inflection points.

This approach extends naturally to mixture distributions. For a mixture $f(x) = \sum_{i=1}^{k} w_i f_i(x)$ with weights $w_i \geq 0$ and $\sum_{i=1}^{k} w_i = 1$, we can construct upper and lower bounds for each component $f_i(x)$ and then linearly combine them. More precisely, we use a weighted sum of the components' bounds (Lemma A.3). To make the implementation efficient, mixture components share the grid, which in turn simplifies combining them by summing up the values of the grid points.

**Theorem 3.1** (**Rejection constant using linear density approximation**). *Let $f_T(x)$ be a target density with piecewise linear upper bound $g_T(x)$ constructed using grid points $\{x_0, x_1, \ldots, x_n\}$ as described above, and let $f_P(x)$ be a proposal density with a lower bound $h_P(x)$, on the same grid. Then, the upper bound on the rejection sampling constant $M$ is given by: $\bar{M} = \max_{i \in \{0,1,\ldots,n\}} \frac{g_T(x_i)}{h_P(x_i)}$.*

*Proof.* Appendix A.1 contains the full proof. We first show that bounding the densities bounds the rejection constant. Then, we show that the linear segments, as described, correctly bound a density function. Finally, we exploit monotonicity of the linear approximation to prove the main claim. □

This is a general approach for finding an upper bound on the true rejection constant which works for many real-world distributions. In Section 3.3 we list some examples. Figure 2 shows an example construction of bounds. The method is particularly effective because it requires evaluating the bounds only at the grid points, making it computationally efficient even for complicated distributions.

The algorithm for finding the rejection constant is described in detail in Appendix A.2. First, Algorithm 2 returns linear segments given the distribution and the grid of points. For that we need to be able to evaluate the density function and its derivative. Then, Algorithm 3 returns the rejection constant given the linear segments. All operations can be computed efficiently, in parallel.

## 3.3 Cataloging common distributions

Recall the exponential PDF is given by $f(x; \lambda) = \lambda e^{-\lambda x}$. We can easily show this function is convex by computing the second derivative $f''(x) = \lambda^3 e^{-\lambda x}$ and noticing this is always positive. In this case, to define the grid for the construction of linear segments, we can choose any arbitrary set of

| Distribution | PDF $f(x)$ | Derivative $f'(x)$ | Inflection points |
|---|---|---|---|
| Exponential | $\lambda e^{-\lambda x}$ | $-\lambda f(x)$ | $\emptyset$ |
| Gamma | $\frac{\beta^\alpha}{\Gamma(\alpha)} x^{\alpha-1} e^{-\beta x}$ | $\left(\frac{\alpha-1}{x} - \beta\right) f(x)$ | $\frac{\alpha-1\pm\sqrt{\alpha-1}}{\beta}$ |
| Log-normal | $\frac{1}{x\sigma\sqrt{2\pi}} e^{-\frac{(\log x - \mu)^2}{2\sigma^2}}$ | $\left(-\frac{1}{x} - \frac{\log x - \mu}{\sigma^2 x}\right) f(x)$ | $e^{\mu + \frac{\sigma^2}{2}\left(-3\pm\sqrt{1+\frac{4}{\sigma^2}}\right)}$ |
| Weibull | $\frac{k}{\lambda}\left(\frac{x}{\lambda}\right)^{k-1} e^{-\left(\frac{x}{\lambda}\right)^k}$ | $\left(\frac{k-1}{x} - \frac{k}{\lambda}\left(\frac{x}{\lambda}\right)^{k-1}\right) f(x)$ | $2^{-\frac{1}{k}}\lambda\left(\frac{-3+3k\pm\sqrt{5k^2-6k+1}}{k}\right)^{\frac{1}{k}}$ |

Table 1: Commonly used distributions in TPP models.

points. However, most densities are not strictly convex so this will not work in general. What we do, instead, is find the intervals on which the density is either concave or convex.

The intervals are bounded by inflection points, which we get by solving $f''(x) = 0$. This is possible to find for most distributions. In Table 1 we show some commonly used distributions, along with their first derivative and inflection points. The full derivation of all the terms is provided in Appendix B.1 for exponential distribution, in B.2 for Gamma, B.3 for log-normal, and in Appendix B.4 for Weibull.

### 3.4 EFFICIENT SAMPLING ALGORITHM

Our proposed method leverages a two-step approach involving renewal sampling and acceptance/rejection to efficiently sample multiple future events.

**Renewal sampling.** The existing encoder has been trained to predict the distribution of the next event based on historical data. Let $\mathcal{H}_i$ denote the history of events up to event time $t_{i-1}$. The encoder represents the sequence with the state $\boldsymbol{h}_i$ and predicts the distribution of the next event $p(\tau_i, x_i|\mathcal{H}_i)$. This prediction serves as a proposal distribution, allowing us to generate multiple future events simultaneously. We denote the set of $l$ proposed future events with $\mathcal{S} = \{(\tau_{i+1}, x_{i+1}), \ldots, (\tau_{i+l}, x_{i+l})\}$.

**Sample acceptance.** We accept proposed samples until we encounter the first event where the proposal and target distributions diverge. The model processes all generated samples $\mathcal{S}$ in parallel, producing hidden states $\boldsymbol{h}_{i+1}, \boldsymbol{h}_{i+2}, \ldots, \boldsymbol{h}_{i+l}$, and target distributions $p^*(\tau_{i+1}, x_{i+j}|\mathcal{H}_{\tau_{i+j}})$, one for each proposed sample $i + j$. Next, we compute the rejection constants $M_{i+j}$ based on the proposal distribution $p(\tau_i, x_i|\mathcal{H}_i)$ and the target distribution $p^*(\tau_{i+j}, x_{i+j}|\mathcal{H}_{i+j})$.

The samples are then independently flagged as accepted or discarded based on the following criterion: for each proposed sample $(\tau_{i+j}, x_{i+j})$, we accept it with probability

$$\mathrm{P}_{i+j} = \frac{p^*(\tau_{i+j}, x_{i+j}|\mathcal{H}_{i+j})}{M_{i+j}p(\tau_i, x_i|\mathcal{H}_i)}.$$

We find the first $j \in \{1, \ldots, l\}$ such that $(i + j + 1)$th event is discarded. We then discard all the events after $i + j$, set the hidden state of the model to $\boldsymbol{h}_{i+j}$ and repeat the above procedure. This results in *exact* sampling from the model. This is easy to show by examining the chain of validity that governs sequential events in TPPs. When sampling from the target distribution, each accepted event depends on having a valid history of previous events. The rejection constant precisely quantifies the maximum ratio between the target density (conditioned on the true, evolving history) and the proposal density (conditioned on the static initial history). By stopping at the first rejection, we ensure that every accepted sequence follows **the exact** conditional structure of the target distribution.

All of the described steps can be computed in parallel. The efficiency in terms of memory requirements will depend on the choice of $l$ and the average rejection rate. Therefore, $l$ can be adjusted on the fly to maximize efficiency. Note that the first event at step $i + 1$ will always be accepted, since the proposal and target distributions are identical. The overhead computations of this method compared to a conventional sampling scheme is in obtaining the constants $M_{i+j}$.

**Batching.** In case of batched samples, we keep track of the shortest sequence in the batch and continue generating events for all sequences until the shortest one is completed. Assuming that the sequences come from the same process, we expect that the rejection rate will be similar over longer simulation horizons. Then, the extra samples that are generated for some sequences are trimmed. An alternative implementation can use packed sequences to dynamically exclude completed sequences.

**Algorithm 1 Efficient event sampling.**

Let's examine a concrete example. If we have historical data up to the current time, we might want to generate 100 future events using speculative step of 3. First, the encoder processes history into a hidden state $\boldsymbol{h}_0$. The decoder outputs proposal distribution $p(\tau_1, x_1|\boldsymbol{h}_0)$. We sample 3 events from this, e.g., "email" at $\tau_1 = 0.5$, "call" at $\tau_2 = 1.7$, and "meeting" at $\tau_3 = 0.2$. Then, for each sample, we compute its updated hidden state (how history would look if it occurred), target distribution $p^\star$, and the rejection constant. After computing the acceptance probability, we might reject the third event and append the sequence with the accepted events. We repeat this process, starting from $\boldsymbol{h}_2$, until we generate all future events.

1: **Input:** Historical data $\mathcal{H}_i$, encoder model Enc, decoder model Dec, rejection constant function RejectionConst, rejection function IsRejected
2: **Output:** Samples $\{(\tau_{i+1}, x_{i+1}), \ldots, (\tau_{i+l}, x_{i+l})\}$
3: $\mathcal{S} \leftarrow \{\}$          # Initialize set of samples
4: $\boldsymbol{h} \leftarrow \mathrm{Enc}(\mathcal{H}_i)$       # History hidden state
5: $p \leftarrow \mathrm{Dec}(\boldsymbol{h})$       # Proposal distribution
6: **while** $|\mathcal{S}| \leq l$ **do**
7:    $(\tau_{i+j}, x_{i+j}) \sim p(\tau_i, x_i|\boldsymbol{h})$    # Proposal events
8:    $\boldsymbol{h}_{i+j} \leftarrow \mathrm{Enc}(\tau_{i+j}, x_{i+j})$    # New states
9:    $p^*_{i+j} \leftarrow \mathrm{Dec}(\boldsymbol{h}_{i+j})$    # Target distributions
10:    $M_{i+j} \leftarrow \mathrm{RejectionConst}(p, p^*_{i+j})$
11:    $u_{i+j} \leftarrow \mathrm{IsRejected}(M_{i+j})$ # Boolean 0-1 output
12:    $k \leftarrow \arg\min_j u_{i+j}$    # Find first rejection
13:    $\mathcal{S} \leftarrow \mathcal{S} \cup \{(\tau_{i+j}, x_{i+j})\}_{j=1}^{k-1}$   # Append samples
14:    $\boldsymbol{h} \leftarrow \boldsymbol{h}_{i+k-1}$    # Update state
15:    $p \leftarrow p^*_{i+k-1}$    # Update proposal
16:    $i \leftarrow k - 1$    # Update index
17: **end while**

Algorithm 1 shows the proposed method. For clarity, wherever $\bullet_{i+j}$ is used, it is implied that all values $\{\bullet_{i+j}\}_{j=1}^l$ are computed in parallel. We split the model into an encoder, which only outputs the hidden states; and the decoder, that outputs distributions. To further enhance the sampling efficiency, we can adopt a non-exact approach. A variant of Algorithm 1, which we evaluate in Section 5, utilizes the $k$th rejection instead of the first. Other non-exact methods may involve different approximations.

## 4 RELATED WORK

Envelope methods (Devroye, 1984; Gilks & Wild, 1992) construct the upper bound on the log-density in order to take samples from the upper bound which are then accepted w.r.t. the true density. On the other hand, we can sample directly from our densities, that is, our method uses bounded functions only to compute the rejection constant. Doing this, we obtain exact sampling and speed up the process by parallelizing the ancestral sampling. Görür & Teh (2011) propose a concave-convex approximation similar to ours, however, our Algorithm 1 is less expensive and readily parallelizable.

Most TPP models are autoregressive: Hawkes process (Hawkes, 1971) defines excitement through adjacency matrix and exponential kernel; RMTPP (Du et al., 2016) uses RNN encoder and a simple distribution for delta times; Neural Hawkes (Mei & Eisner, 2017) combines RNNs with Hawkes intensity; and intensity-free models (Shchur et al., 2020) generalize to any distribution. Other research has explored modifications to the encoder (Zuo et al., 2020; Chen et al., 2018); and decoder, incorporating different intensity functions (Omi et al., 2019; Lüdke et al., 2023).

Xue et al. (2024) present a benchmark for TPPs that consolidates existing models and datasets. Their findings reveal that neural models outperform classical TPPs, with small performance variation among different models. Notably, intensity-free models still achieve state-of-the-art results. Karpukhin et al. (2024) suggest that predicting a full window is a strong baseline for long-horizon forecasting, however, autoregressive models remain the dominant paradigm in literature and practice. Xue et al. (2022) enhance long-horizon predictions through a hybrid model that combines an autoregressive base with an energy function for reweighing. In contrast, our work focuses on improving sampling efficiency; it can be combined with this method to enable parallel sampling of multiple next events.

Speculative decoding (Stern et al., 2018) has resurfaced in LLMs as a way to accelerate sampling (Qi et al., 2020; Gloeckle et al., 2024). LLM training and sampling is similar to TPP models, and by extension to other autoregressive models, like those from time series forecasting. Our method can be directly applied to text generation, however, it is not suitable for this task since in language domain, consecutive token distributions vary considerably. So, while the two approaches share similarities, a key distinction is that we do not require learning to predict multiple next steps, allowing us to apply our method to existing models without retraining. This is not possible to achieve in the text domain.

**10 dimensions**

| Sparsity \ $A_{\max}$ | 0.05 | 0.1 | 0.2 | 0.5 | 1.0 |
|---|---|---|---|---|---|
| 0.1 | 4.70 | 4.37 | 4.30 | 4.57 | 4.00 |
| 0.2 | 4.80 | 4.17 | 3.80 | 3.97 | 4.33 |
| 0.3 | 4.83 | 4.30 | 3.77 | 4.30 | 4.07 |
| 0.4 | 4.57 | 4.47 | 3.30 | 4.37 | 3.73 |
| 0.5 | 4.93 | 4.43 | 3.93 | 3.47 | 3.97 |
| 0.6 | 4.87 | 4.63 | 3.93 | 3.57 | 3.03 |
| 0.7 | 4.97 | 4.80 | 4.47 | 2.77 | 3.60 |
| 0.8 | 4.80 | 4.77 | 4.67 | 3.80 | 3.77 |
| 0.9 | 5.00 | 4.97 | 4.57 | 3.90 | 3.10 |

**20 dimensions**

| Sparsity \ $A_{\max}$ | 0.05 | 0.1 | 0.2 | 0.5 | 1.0 |
|---|---|---|---|---|---|
| 0.1 | 4.20 | 4.10 | 4.03 | 4.13 | 4.47 |
| 0.2 | 4.57 | 4.03 | 4.63 | 4.07 | 4.07 |
| 0.3 | 4.77 | 4.03 | 3.97 | 4.13 | 3.47 |
| 0.4 | 4.60 | 3.90 | 3.53 | 3.80 | 4.23 |
| 0.5 | 4.93 | 4.23 | 3.23 | 3.50 | 3.63 |
| 0.6 | 4.80 | 4.20 | 3.23 | 3.27 | 3.83 |
| 0.7 | 4.90 | 4.43 | 3.97 | 2.60 | 2.97 |
| 0.8 | 4.97 | 4.87 | 3.73 | 2.60 | 2.57 |
| 0.9 | 5.00 | 4.90 | 4.00 | 3.03 | 2.13 |

**40 dimensions**

| Sparsity \ $A_{\max}$ | 0.05 | 0.1 | 0.2 | 0.5 | 1.0 |
|---|---|---|---|---|---|
| 0.1 | 3.83 | 3.60 | 4.33 | 3.87 | 4.03 |
| 0.2 | 3.70 | 3.40 | 3.73 | 3.80 | 3.77 |
| 0.3 | 3.60 | 3.90 | 3.80 | 3.80 | 3.77 |
| 0.4 | 3.93 | 3.93 | 3.10 | 2.93 | 3.90 |
| 0.5 | 4.67 | 4.03 | 3.43 | 3.83 | 3.43 |
| 0.6 | 4.67 | 2.53 | 2.70 | 3.57 | 3.07 |
| 0.7 | 4.70 | 3.53 | 2.60 | 2.63 | 2.57 |
| 0.8 | 4.80 | 4.30 | 2.37 | 2.43 | 2.03 |
| 0.9 | 5.00 | 4.73 | 3.80 | 2.13 | 1.87 |

**80 dimensions**

| Sparsity \ $A_{\max}$ | 0.05 | 0.1 | 0.2 | 0.5 | 1.0 |
|---|---|---|---|---|---|
| 0.1 | 3.53 | 4.00 | 3.57 | 3.67 | 3.87 |
| 0.2 | 3.40 | 4.07 | 2.97 | 3.57 | 3.93 |
| 0.3 | 3.60 | 3.43 | 3.60 | 3.00 | 3.07 |
| 0.4 | 3.67 | 3.37 | 3.83 | 3.67 | 3.03 |
| 0.5 | 3.13 | 2.80 | 3.33 | 3.33 | 2.57 |
| 0.6 | 3.47 | 3.17 | 2.93 | 3.30 | 3.37 |
| 0.7 | 3.57 | 2.30 | 1.97 | 2.90 | 2.73 |
| 0.8 | 4.67 | 2.67 | 1.90 | 2.47 | 2.30 |
| 0.9 | 4.63 | 4.17 | 2.87 | 1.67 | 1.30 |

Figure 3: Average accepted step (out of 5 proposed events) for different configurations of multivariate Hawkes process. Lower sparsity and larger adjacency values define stronger mark interactions leading to lower acceptance ratio, nevertheless, all configurations show good acceptance ratios.

## 5 EXPERIMENTS

In Section 5.1 we motivate our method with two synthetic examples which have strong connection to real world problems. Section 5.2 shows that real-world data is not-stationary meaning our results in Section 5.3 cannot be replicated with a simpler underlying model. Section 5.4 shows the empirical runtime improvements. Finally, in Section 5.5 we show a study on a real-world problem in finance.

In our study we analyze seven standard event sequence datasets: Amazon (Ni et al., 2019), Earthquake (EQ) (Xue et al., 2024), Retweet (Zhou et al., 2013), Stack Overflow (SO) (Leskovec & Sosič, 2016), Taobao (Xue et al., 2022), and Taxi (Whong, 2014), processed by Xue et al. (2024); and Reddit (Kumar et al., 2018). These datasets have different properties and are a good representation of the real-world data. Sequences can have hundreds of events and mark dimensions range from single digit to 985. Data is described in detail in Appendix C, see e.g., Table 4 and Figure 6.

### 5.1 MOTIVATING EXAMPLES

**Multivariate Hawkes Process.** Marks in a TPP may or may not interact with each other, leading to different sparsity of the adjacency matrix. For instance, in a social network, messages can be treated as events, and not all users have to be connected. To simulate this we generate data from a multivariate Hawkes process (Hawkes, 1971; Bacry et al., 2017). Adjacency matrix is uniformly sampled with values from 0 to $A_{\max}$ and a percentage of values is set to 0. We vary sparsity from 10% to 90%, dimension from 10 to 80, $A_{\max}$ from 0.05 to 1, and the decay factor is either 0.2 or 1.

A single experiment configuration defines a Hawkes process taking the values from the above ranges. We use the true intensity function to compute the rejection constants, acceptance rates, and average accepted step size. We use a maximum speculative step size of 5 and average the results over 30 runs.

Figure 3 shows the average accepted step. We can see that increasing the dimension, connectedness and adjacency strength decreases the acceptance ratio. Crucially, we show that even with a high dimension and low sparsity we obtain good acceptance rates. This confirms the practical utility of our method for diverse TPPs and gives us confidence that it can be applied in real-world tasks.

**Jump Process.** We consider a process that cycles through periods of different constant intensities. For instance, server logs exhibit such behavior; some jobs print at a constant rate, but we can have multiple jobs that start and end at random times, each having its own intensity. This can be described by a jump process: behaving as a renewal process most of the time but experiencing random jumps in intensity function. We construct the data by stitching together sequences from different homogeneous processes by first sampling interval durations and then sampling random intensities which finally generate events on the interval. We use GRU (Cho et al., 2014) with an exponential distribution.

Given initial sequences, we generate 100 new realizations with length 3000. Figure 8 in Appendix D.2 illustrates different samples from the model based on one input sequence, demonstrating the model's ability to generate long intervals of constant intensity with sudden changes. The acceptance rate on the whole test data remains consistently around 90%, with an average accepted step length of 13.

## 5.2 REAL DATA IS USUALLY NON-STATIONARY

If the data originates from a stationary process, we could learn a single distribution for the next event, disregarding all historical context, which would automatically simplify sampling. This would be the same as learning a renewal TPP model. We test this hypothesis by evaluating whether incorporating history significantly enhances modeling quality. We consider three scenarios: (1) full history—using the canonical TPP with complete past context; (2) Markov—utilizing only the most recent event to predict the next; and (3) no history—a TPP with a stationary distribution unaffected by past events.

The training setup is explained in Appendix D.3. Table 6 shows the likelihood, mark accuracy, and time RMSE results averaged over five runs. We include the most common class prediction to demonstrate that models without full history mainly learn the marginal distribution. Full history model is clearly beneficial, indicating that real data is predominantly non-stationary. Consequently, an alternative to our method cannot be a renewal process, as it would significantly reduce performance.

## 5.3 LONG-HORIZON SAMPLING ON BENCHMARK DATA

We use standard benchmark datasets as described at the start of Section 5. Our model has a GRU encoder with 256 hidden dimension. The decoder is a log-normal mixture distribution with 32 components. After training is completed, the models are used to generate new realizations starting from the initial sequences that are given in the held-out test set. We generate the sequences using regular one-by-one sampling which is the *ground truth*, and compare it to speculative sampling using our linear estimation of the rejection constant. We additionally do speculative sampling with Monte Carlo estimation of the rejection constant, the results are discussed in Appendix D.4.

For each sequence, we take 10 samples, each consisting of 100 events, and use a speculative step of 5. We also implement approximate schemes for further improvements in sampling efficiency, namely, top-k sampling where we accept all events up to the $k$th event flagged for rejection. We measure the average acceptance step and various distances between empirical distributions of events from the conventional and speculative sampling. We report KL-divergence between marks, maximum mean discrepancy of arrival times, and log-likelihood ratio; all formally defined in Appendix D.5.

Table 2 shows the distances between the true sampling distribution and the speculative samples. The results demonstrate that large speculative steps are achieved for most datasets without compromising sampling quality. The only exception is the Taxi dataset, which has specific structure where marks alternate between two values, i.e., mark 1 always follows mark 2 and vice versa, limiting the effectiveness of our approach in this specific case. This results in large rejection rates of the categorical distribution but it can be fixed by augmenting the proposal distribution. Additional results with error bars and other metrics are in Table 8. For results using different encoders, such as transformer and convolutional neural network, see Appendix D.6.

## 5.4 RUNTIME IMPROVEMENTS

Having demonstrated that speculative sampling generates equivalent samples to the true process while accepting multiple events per iteration, we now quantify the computational benefits through wall-clock time measurements (with hardware specifications in Appendix D.1). We break down the sampling procedure into its key components as outlined in Algorithm 1. For instance, "Encoder" measurement represents the total time spent on encoder computations.

Table 3 presents the timing comparison, additional detailed results are available in Appendix D.6. The measurements demonstrate substantial speedup, particularly for datasets with higher acceptance rates.

Table 2: Quality of samples and average accepted step for different top-k, compared to ground truth.

| | MMD | | | | KL-divergence | | | | Log-likelihood ratio | | | | Step | | |
|---|---|---|---|---|---|---|---|---|---|---|---|---|---|---|---|
| | True | Top-1 | Top-2 | Top-3 | True | Top-1 | Top-2 | Top-3 | True | Top-1 | Top-2 | Top-3 | Top-1 | Top-2 | Top-3 |
| Amazon | 0.2 | 0.2 | 0.19 | 0.2 | 7.26 | 7.33 | 7.26 | 7.28 | 0.02 | -0.02 | -0.03 | -0.05 | 2.8905 | 5.9309 | 8.8409 |
| EQ | 0.19 | 0.2 | 0.19 | 0.18 | 3.9 | 3.98 | 3.99 | 3.86 | 0.04 | -0.13 | -0.05 | 0.03 | 3.0387 | 6.1882 | 9.1424 |
| Reddit | 0.19 | 0.19 | 0.19 | 0.2 | 6.36 | 6.4 | 6.45 | 6.26 | -0.06 | -0.15 | -0.16 | -0.21 | 2.1848 | 4.3222 | 6.5014 |
| Retweet | 0.2 | 0.19 | 0.19 | 0.19 | 1.89 | 1.92 | 1.87 | 1.99 | 0.02 | 0.02 | 0.01 | -0.03 | 3.7707 | 8.2337 | 12.6512 |
| SO | 0.19 | 0.18 | 0.19 | 0.18 | 6.93 | 6.97 | 6.93 | 7.0 | 0.01 | -0.01 | -0.06 | -0.07 | 2.1695 | 4.282 | 6.225 |
| Taobao | 0.2 | 0.2 | 0.2 | 0.2 | 8.33 | 8.59 | 8.34 | 8.63 | 0.02 | -0.07 | -0.21 | -0.24 | 1.7783 | 3.4225 | 5.0025 |
| Taxi | 0.19 | 0.19 | 0.21 | 0.21 | 2.64 | 2.71 | 6.38 | 6.27 | 0.02 | 0.02 | 2.68 | 2.81 | 1.0003 | 2.1283 | 3.0359 |

Table 3: Average total time (in ms) for speculative sampling, compared to a conventional method.

| | Encoder (process history) | | | | Decoder (output distributions) | | | | Sample next event | | | |
|---|---|---|---|---|---|---|---|---|---|---|---|---|
| | True | Top-1 | Top-2 | Top-3 | True | Top-1 | Top-2 | Top-3 | True | Top-1 | Top-2 | Top-3 |
| Amazon | 36.36 | 16.66 | 11.4 | 7.47 | 42.04 | 37.04 | 23.11 | 12.87 | 47.84 | 12.81 | 8.15 | 4.74 |
| EQ | 46.53 | 20.61 | 10.99 | 7.82 | 54.09 | 45.83 | 21.32 | 13.4 | 62.69 | 16.11 | 7.78 | 5.09 |
| Reddit | 37.89 | 25.05 | 19.2 | 13.81 | 43.41 | 55.34 | 36.2 | 22.94 | 49.53 | 19.22 | 13.54 | 9.1 |
| Retweet | 39.24 | 13.81 | 7.73 | 5.94 | 44.13 | 29.31 | 14.11 | 9.4 | 50.35 | 10.27 | 5.1 | 3.53 |
| SO | 49.02 | 29.85 | 17.36 | 13.13 | 50.61 | 59.83 | 30.87 | 21.46 | 60.72 | 22.35 | 12.3 | 8.88 |
| Taobao | 49.06 | 37.56 | 22.12 | 16.22 | 50.71 | 76.34 | 40.45 | 27.12 | 60.79 | 28.12 | 15.71 | 10.99 |
| Taxi | 49.18 | 54.33 | 29.86 | 22.62 | 51.13 | 111.12 | 55.5 | 38.98 | 61.03 | 40.71 | 21.7 | 15.61 |

Figure 4: Limit order book samples. (Left) Samples generated with a conventional autoregressive method. (Right) Samples generated with a speculative sampling method.

This performance gain stems from more efficient utilization of parallel computing capabilities in modern hardware. Since neural network operations on batched data incur similar overhead regardless of batch sizes, our approach achieves better throughput by reducing the total number of separate calls.

The rejection constant runtime depends on the implementation and exact parameters. The current implementation prioritizes clarity and didactic value rather than computational efficiency, while other modules leverage optimized native operations. Despite this, we achieve significant overall speedup.

### 5.5 REAL-WORLD APPLICATION: LIMIT ORDER BOOKS

Limit order book (LOB) records outstanding buy and sell orders for an asset, organized by price level. Real-time messages, such as new orders or cancellations, are processed based on price and time priority, ensuring efficient trade execution. Messages can be modeled with a TPP, where individual order timings are arrival times and their contents are marks. Accurately modeling LOB messages is essential for enhancing trading strategies. We use publicly available data for the MSFT symbol from a single day,[1] comprising 600k messages with various features, detailed in Appendix C.2. The data is divided into training sequences of length 200, with four mark types.

A unique challenge arises as messages can arrive simultaneously which occurs in about 8% of the data, conflicting with the assumptions of a simple point process. To address this, we quantize delta times using quantile bins, allowing for zero values. The model then predicts one categorical distribution for inter-arrival times and another categorical distribution for marks. We simulate 100 future steps 10 times, achieving an average speculative step size of 2.82, and 5.19 for top-2 sampling. Figure 4 shows empirical samples using different sampling methods. In Appendix D.7 we show some *stylized facts* that demonstrate that the samples generated with a speculative scheme have the same properties as the conventional sampling. Figure 9 shows transition matrices and Figure 10 cumulative counts.

## 6 DISCUSSION

In this paper, we introduce a novel method that improves the simulation efficiency of TPPs by enabling parallel sampling of multiple next events, which is crucial for rapid event sequence generation. Notably, our method is effective even with non-renewal data, allowing for broad applicability without altering existing models. Our approach utilizes envelope approximation, which can be applied to a wide range of functions, including most common distribution choices and their mixtures. Our approach can be adapted to various model definitions, and we offer implementations for most common choices. Future work might explore better proposal distributions to further boost efficiency.

---

[1] https://lobsterdata.com/info/DataSamples.php

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

# A   THEORETICAL RESULTS

## A.1   PROOF OF THEOREM 3.1

First, we show some results that will help us prove the main claim.

**Lemma A.1.** *Let $g_T(x)$ be the upper bound of the target density $f_T(x)$, and let $h_P(x)$ be the lower bound of the proposal density $f_P(x)$. Then the ratio $\tilde{M} = \max_x \frac{g_T(x)}{h_P(x)}$ is the upper bound of the rejection sampling constant $M = \max_x \frac{f_T(x)}{f_P(x)}$.*

*Proof.* Since $f_T(x) \leq g_T(x)$ and $h_P(x) \leq f_P(x)$, we know $\frac{f_T(x)}{f_P(x)} \leq \frac{g_T(x)}{h_P(x)}$, which finally gives $M = \max_x \frac{f_T(x)}{f_P(x)} \leq \max_x \frac{g_T(x)}{h_P(x)} = \tilde{M}$. $\qquad\square$

**Lemma A.2.** *For any density function $f(x)$ and grid $\{x_0, x_1, \ldots, x_n\}$, $f$ can be approximated by a piecewise linear function where, in convex regions $[x_i, x_{i+1}]$, $f$ is upper-bounded by the line segment connecting $(x_i, f(x_i))$ and $(x_{i+1}, f(x_{i+1}))$ and lower-bounded by the line with slope $f'(x_m)$ at $x_m = \frac{x_i + x_{i+1}}{2}$ passing through $(x_m, f(x_m))$. In concave regions, these bounds are reversed. Any function can be bounded by decomposing its domain into convex and concave regions.*

*Proof.* We proceed by establishing bounds for convex and concave regions separately, then show how these can be combined.

**Case 1: Convex regions.** Let $[x_i, x_{i+1}]$ be an interval where $f$ is convex, and let $x_m = \frac{x_i + x_{i+1}}{2}$.

For the upper bound, by definition of convexity, for any $\lambda \in [0, 1]$ and $x = \lambda x_i + (1 - \lambda)x_{i+1}$:

$$f(x) \leq \lambda f(x_i) + (1 - \lambda)f(x_{i+1})$$

This is precisely the line segment connecting $(x_i, f(x_i))$ and $(x_{i+1}, f(x_{i+1}))$, confirming it as an upper bound.

For the lower bound, let $L(x)$ be the line with slope $f'(x_m)$ passing through $(x_m, f(x_m))$:

$$L(x) = f(x_m) + f'(x_m)(x - x_m)$$

For a convex function, any tangent line lies below the function, thus:

$$f(x) \geq f(x_m) + f'(x_m)(x - x_m) = L(x) \quad \forall x$$

**Case 2: Concave regions.** For intervals where $f$ is concave, the inequalities reverse. The line segment becomes a lower bound, and the tangent line at $x_m$ becomes an upper bound, by the same reasoning applied to $-f$.

**Combining regions:** Any continuous function can be decomposed into intervals where it is either convex or concave, assuming $f''$ exists and changes sign a finite number of times in any bounded interval. This holds for all the densities of interest. By applying the appropriate bounds in each region, we obtain piecewise linear upper and lower bounds for the entire function. $\qquad\square$

This construction assumes $f$ is twice differentiable almost everywhere to identify convex and concave regions. We also assume that $f'(x_m)$ exists at each midpoint. The approximation error depends on the grid spacing and the maximum curvature of $f$. Note that the endpoints from two adjacent linear segments do not have to meet in the same point.

**Lemma A.3.** *Let $f(x) = \sum_{i=1}^{k} w_i f_i(x)$ be a mixture of density functions with weights $w_i \geq 0$ and $\sum_{i=1}^{k} w_i = 1$. Let $g_i(x)$ be a linear upper bound and $h_i(x)$ a lower bound of a component $f_i(x)$ according to Lemma A.2. Then, $g(x) = \sum_{i=1}^{k} w_i g_i(x)$ is an upper bound of $f(x)$, and $h(x) = \sum_{i=1}^{k} w_i h_i(x)$ is a lower bound of $f(x)$.*

*Proof.* Since $g_i(x) \geq f_i(x)$ for all $x$ and for each $i \in \{1, 2, \ldots, k\}$, and $w_i \geq 0$, we have:

$$w_i g_i(x) \geq w_i f_i(x) \quad \forall x, \forall i \tag{3}$$

Summing over all components:

$$\sum_{i=1}^{k} w_i g_i(x) \geq \sum_{i=1}^{k} w_i f_i(x) \tag{4}$$

$$g(x) \geq f(x) \tag{5}$$

Similarly, since $h_i(x) \leq f_i(x)$ for all $x$ and for each $i \in \{1, 2, \ldots, k\}$, we have:

$$\sum_{i=1}^{k} w_i h_i(x) \leq \sum_{i=1}^{k} w_i f_i(x) \tag{6}$$

$$h(x) \leq f(x) \tag{7}$$

Since each $g_i(x)$ and $h_i(x)$ is piecewise linear by construction, and a weighted sum of piecewise linear functions remains piecewise linear, both $g(x)$ and $h(x)$ maintain the piecewise linear structure. $\square$

Lemma A.3 holds even if component functions are approximated on different grids. In that case, one has to take the union over all the grid points to get the final shape. A simpler and computationally friendly approach is to predefine the grid that will share the convexity for all the components. Finally, having proven Lemmas A.1, A.2, and A.3, we can prove Theorem 3.1 from the main text.

*Proof.* (**Theorem 3.1**) Between any two adjacent grid points $[x_i, x_{i+1}]$, both $g_T(x)$ and $h_P(x)$ are linear functions:

$$g_T(x) = a_i x + b_i \tag{8}$$

$$h_P(x) = c_i x + d_i \tag{9}$$

where $a_i, b_i, c_i, d_i$ are constants determined by the respective bounds.

The ratio between grid points is therefore:

$$r(x) = \frac{g_T(x)}{h_P(x)} = \frac{a_i x + b_i}{c_i x + d_i} \tag{10}$$

This ratio function $r(x)$ is either:

1. Constant (when $\frac{a_i}{c_i} = \frac{b_i}{d_i}$), in which case the maximum is attained everywhere, including at the endpoints,

2. Strictly monotonic (either increasing or decreasing), as the derivative $r'(x) = \frac{a_i d_i - b_i c_i}{(c_i x + d_i)^2}$ has constant sign. In this case, the maximum in $[x_i, x_{i+1}]$ must occur at either $x_i$ or $x_{i+1}$.

Since this holds for all intervals $[x_i, x_{i+1}]$, the global maximum of $r(x)$ must occur at one of the grid points $\{x_0, x_1, \ldots, x_n\}$. Therefore:

$$\tilde{M} = \max_x r(x) = \max_{i \in \{0, 1, \ldots, n\}} \frac{g_T(x_i)}{h_P(x_i)} \tag{11}$$

$\square$

Theorem 3.1 shows us how to construct the grid made out of piecewise linear segments. Since it works for mixture distributions, e.g., with components defined in Section 3.3, and since mixture distributions build a universal approximator for TPPs (Shchur et al., 2020), our method can be considered a universal approach for speculative sampling.

## A.2 ALGORITHM FOR FINDING THE REJECTION CONSTANT

Algorithm 2 shows us how to get the upper and lower bound for any density, as long as we can evaluate this density and its derivative in any point. Algorithm 3 shows us how to compute the rejection constant given the target and proposal density.

---

**Algorithm 2** GetBounds function for bounding density

---

1: **Input:** Distribution $p$, left segment bounds $\mathcal{X}_{\text{left}}$, right segment bounds $\mathcal{X}_{\text{right}}$, boolean upperBound
2: **Output:** Left and right segment values which define the linear segments which bound the density
3: $\bar{\mathcal{X}} \leftarrow (\mathcal{X}_{\text{left}} + \mathcal{X}_{\text{right}})/2$        # Mid points
4: $\bar{\mathcal{P}} \leftarrow p(\bar{\mathcal{X}}),\ \mathcal{P}_{\text{left}} \leftarrow p(\mathcal{X}_{\text{left}}),\ \mathcal{P}_{\text{right}} \leftarrow p(\mathcal{X}_{\text{right}})$
5: $\bar{\mathcal{P}}' \leftarrow p'(\bar{\mathcal{X}})$        # Derivative in mid points
6: $\mathcal{Z}_{\text{left}} \leftarrow \bar{\mathcal{P}}'(\mathcal{X}_{\text{left}} - \bar{\mathcal{X}}) + \bar{\mathcal{P}}$        # Tangent values in edges
7: $\mathcal{Z}_{\text{right}} \leftarrow \bar{\mathcal{P}}'(\mathcal{X}_{\text{right}} - \bar{\mathcal{X}}) + \bar{\mathcal{P}}$
8: **if** upperBound **then**
9:     $\mathcal{Y}_{\text{left}}, \mathcal{Y}_{\text{right}} \leftarrow \max(\mathcal{P}_{\text{left}}, \mathcal{Z}_{\text{left}}), \max(\mathcal{P}_{\text{right}}, \mathcal{Z}_{\text{right}})$
10: **else**
11:     $\mathcal{Y}_{\text{left}}, \mathcal{Y}_{\text{right}} \leftarrow \min(\mathcal{P}_{\text{left}}, \mathcal{Z}_{\text{left}}), \min(\mathcal{P}_{\text{right}}, \mathcal{Z}_{\text{right}})$
12: **end if**
13: **Return** $\mathcal{Y}_{\text{left}}, \mathcal{Y}_{\text{right}}$

---

---

**Algorithm 3** RejectionConst for any pair of densities

---

1: **Input:** Proposal distribution $q$, target distribution $p$, target percentile $\alpha$, number of grid points $n$, grid generator GetGridPoints, GetBounds (Algorithm 2)
2: **Output:** Rejection constant $M$
3: $\mathcal{X}_{\text{left}}, \mathcal{X}_{\text{right}} \leftarrow$ GetGridPoints$(q, p, \alpha, n)$
4: $\mathcal{P}_{\text{left}}, \mathcal{P}_{\text{right}} \leftarrow$ GetBounds$(p, \mathcal{X}_{\text{left}}, \mathcal{X}_{\text{right}}, \text{True})$
5: $\mathcal{Q}_{\text{left}}, \mathcal{Q}_{\text{right}} \leftarrow$ GetBounds$(q, \mathcal{X}_{\text{left}}, \mathcal{X}_{\text{right}}, \text{False})$
6: $\mathcal{R} \leftarrow [\mathcal{P}_{\text{left}}/\mathcal{Q}_{\text{left}}, \mathcal{P}_{\text{right}}/\mathcal{Q}_{\text{right}}]$
7: $M \leftarrow \max(\mathcal{R})$
8: **Return** $M$

---

## A.3 IMPROVED REJECTION SAMPLING FOR CATEGORICAL DISTRIBUTION WITH BOUNDED ERROR

For categorical distributions $p_T$ and $p_P$ over a discrete set $\mathcal{X}$, the rejection sampling constant is defined in Equation 2. The acceptance probability is then $\frac{1}{M}$, meaning that on average, $M$ samples must be drawn from $p_P$ to obtain one sample from $p_T$. We can trade-off accuracy for efficiency by defining the $\delta$-truncated rejection constant as:

$$M_\delta = \min\left\{ M \geq 1 \,\Big|\, \sum_{x \in S(M)} p_T(x) \geq 1 - \delta \right\}, \tag{12}$$

where $S(M) = \{x \in \mathcal{X} \mid M \cdot p_P(x) \geq p_T(x)\}$ is the set of elements that satisfy the rejection criterion with constant $M$. To compute $M_\delta$:

1. Calculate ratios $r(x) = \frac{p_T(x)}{p_P(x)}$ for all $x \in \mathcal{X}$,

2. Sort ratios in descending order: $r_1 \geq r_2 \geq \ldots \geq r_n$,

3. Compute cumulative mass of the target distribution: $C_i = \sum_{j=1}^{i} p_T(\hat{x}_j)$, where $\hat{x}_j$ is the element with the $j$th highest ratio,

4. Find the smallest index $i^*$ such that $C_{i^*} \geq \delta$,

5. Set $M_\delta = r_{i^*+1}$.

This corresponds to excluding the elements with the highest ratios, whose collective probability under $p$ is at most $\delta$. When $\delta$ is small, we expect this to have no effect on sampling quality. To quantify the error, we measure the total variation distance: $\text{TV}(\tilde{p}, p) = \frac{1}{2} \sum_{x \in \mathcal{X}} |\tilde{p}(x) - p(x)|$, where $\tilde{p}$ is the distribution resulting from $\delta$-truncated rejection sampling. We show that this error is bounded by $\delta$.

**Lemma A.4.** *Let $\tilde{p}$ be the distribution of samples generated by $\delta$-truncated rejection sampling. The total variation distance between $\tilde{p}$ and the target distribution $p$ is bounded by $TV(\tilde{p}, p) \leq \delta$.*

*Proof.* Let $E = \left\{ x \in \mathcal{X} \mid \frac{p(x)}{q(x)} > M_\delta \right\}$ be the set of excluded elements. Then, by construction, $\sum_{x \in E} p(x) \leq \delta$. The $\delta$-truncated algorithm effectively samples from a re-normalized distribution:

$$\tilde{p}(x) = \begin{cases} \frac{p(x)}{1 - \sum_{y \in E} p(y)} & \text{if } x \notin E \\ 0 & \text{if } x \in E. \end{cases}$$

The total variation distance is bounded by:

$$\text{TV}(\tilde{p}, p) = \frac{1}{2} \sum_{x \in \mathcal{X}} |\tilde{p}(x) - p(x)| = \frac{1}{2} \left( \sum_{x \in E} p(x) + \sum_{x \notin E} \left| \frac{p(x)}{1 - \sum_{y \in E} p(y)} - p(x) \right| \right),$$

which simplifies to $\text{TV}(\tilde{p}, p) = \sum_{x \in E} p(x) \leq \delta$. $\qquad\qquad\square$

A more precise estimation of the error accounts for the partial coverage of excluded categories:

$$\text{TV}_{\text{effective}}(\tilde{p}_T, p_T) = \sum_{x \in E} p_T(x) \left( 1 - \min \left( 1, \frac{M_\delta p_P(x)}{p_T(x)} \right) \right). \tag{13}$$

This represents the fact that categories in $E$ aren't completely unrepresented, they are sampled from $p_P$ and accepted with probability $\frac{M_\delta p_P(x)}{p_T(x)}$, which provides partial coverage.

# B    DISTRIBUTIONS

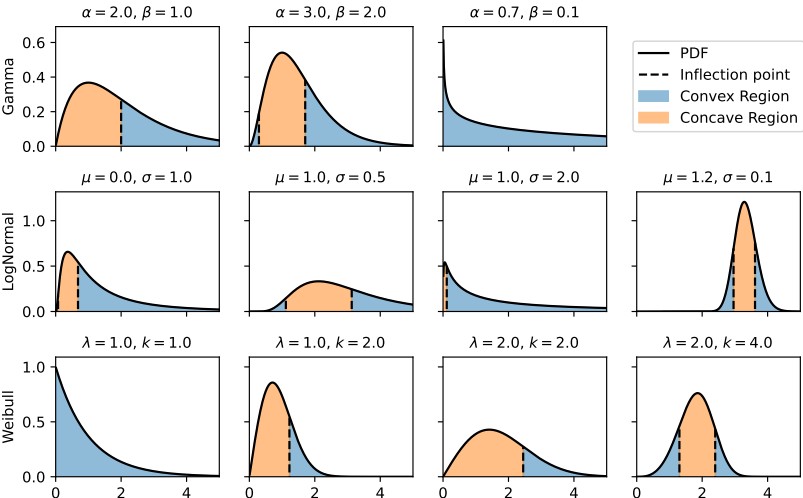

Figure 5: Convexity of Gamma, log-normal and Weibull distributions for different parameters. The first and second derivative, and inflection points are available in closed-form, see Table 1.

## B.1 EXPONENTIAL

The exponential PDF is given by $f(x; \lambda) = \lambda e^{-\lambda x}$, $\lambda > 0$. The first derivative of $f(x)$ is:

$$f'(x) = \frac{d}{dx}\left(\lambda e^{-\lambda x}\right) = -\lambda^2 e^{-\lambda x}.$$

Using this, the second derivative of $f(x)$ is:

$$f''(x) = \frac{d}{dx}\left(-\lambda^2 e^{-\lambda x}\right) = \lambda^3 e^{-\lambda x}.$$

Since $\lambda > 0$ this is always positive, exponential PDF is convex, therefore, we do not have any inflection points.

## B.2 GAMMA

The Gamma PDF is given by $f(x; \alpha, \beta) = \frac{\beta^\alpha}{\Gamma(\alpha)} x^{\alpha-1} e^{-\beta x}$; $\alpha, \beta > 0$. The first derivative of $f(x)$ is:

$$\begin{aligned}
f'(x) &= \frac{\beta^\alpha}{\Gamma(\alpha)} \frac{d}{dx}\left(x^{\alpha-1} e^{-\beta x}\right) \\
&= \frac{\beta^\alpha}{\Gamma(\alpha)}\left[\frac{d}{dx}\left(x^{\alpha-1}\right) e^{-\beta x} + x^{\alpha-1}\frac{d}{dx}\left(e^{-\beta x}\right)\right] \\
&= \frac{\beta^\alpha}{\Gamma(\alpha)}\left[(\alpha-1)x^{\alpha-2}e^{-\beta x} - \beta x^{\alpha-1}e^{-\beta x}\right] \\
&= f(x)\left(\frac{\alpha-1}{x} - \beta\right).
\end{aligned}$$

Then, the second derivative of $f(x)$ is:

$$\begin{aligned}
f''(x) &= f'(x)\left(\frac{\alpha-1}{x} - \beta\right) + f(x)\frac{d}{dx}\left(\frac{\alpha-1}{x} - \beta\right) \\
&= f(x)\left[\left(\frac{\alpha-1}{x} - \beta\right)^2 - \frac{\alpha-1}{x^2}\right]
\end{aligned}$$

We find inflection points by solving $f''(x) = 0$. Since $f(x) > 0$, this simplifies to:

$$\left(\frac{\alpha-1}{x} - \beta\right)^2 = \frac{\alpha-1}{x^2}$$

$$\left|\frac{\alpha-1}{x} - \beta\right| = \sqrt{\frac{\alpha-1}{x^2}}$$

We distinguish two cases:

1. $\frac{\alpha-1}{x} - \beta = \sqrt{\frac{\alpha-1}{x^2}}$

2. $\frac{\alpha-1}{x} - \beta = -\sqrt{\frac{\alpha-1}{x^2}}$

which finally gives us our inflection points:

$$x_1 = \frac{\alpha - 1 - \sqrt{\alpha-1}}{\beta}, \quad x_2 = \frac{\alpha - 1 + \sqrt{\alpha-1}}{\beta}.$$

## B.3 LOG-NORMAL

The log-normal PDF is given by:

$$f(x; \mu, \sigma) = \frac{1}{x\sigma\sqrt{2\pi}}\exp\left(-\frac{(\log x - \mu)^2}{2\sigma^2}\right), \quad \sigma > 0$$

The first derivative of $f(x)$ is:

$$f'(x) = \frac{d}{dx}\left(\frac{1}{x\sigma\sqrt{2\pi}}\exp\left(-\frac{(\log x - \mu)^2}{2\sigma^2}\right)\right)$$

$$= \frac{d}{dx}\left(\frac{1}{x\sigma\sqrt{2\pi}}\right)\exp\left(-\frac{(\log x - \mu)^2}{2\sigma^2}\right) + \frac{1}{x\sigma\sqrt{2\pi}}\frac{d}{dx}\left(\exp\left(-\frac{(\log x - \mu)^2}{2\sigma^2}\right)\right)$$

$$= -\frac{1}{x^2\sigma\sqrt{2\pi}}\exp\left(-\frac{(\log x - \mu)^2}{2\sigma^2}\right) + \frac{1}{x\sigma\sqrt{2\pi}}\exp\left(-\frac{(\log x - \mu)^2}{2\sigma^2}\right)\left(-\frac{1}{\sigma^2}\frac{\log x - \mu}{x}\right)$$

$$= f(x)\left(-\frac{1}{x} - \frac{\log x - \mu}{\sigma^2 x}\right).$$

The second derivative of $f(x)$ is:

$$f''(x) = \frac{d}{dx}\left(f(x)\left(-\frac{1}{x} - \frac{\log x - \mu}{\sigma^2 x}\right)\right)$$

$$= f'(x)\left(-\frac{1}{x} - \frac{\log x - \mu}{\sigma^2 x}\right) + f(x)\frac{d}{dx}\left(-\frac{1}{x} - \frac{\log x - \mu}{\sigma^2 x}\right)$$

$$= f(x)\left(-\frac{1}{x} - \frac{\log x - \mu}{\sigma^2 x}\right)^2 + f(x)\left[\frac{1}{x^2} - \frac{1}{\sigma^2}\frac{d}{dx}\left(\frac{\log x - \mu}{x}\right)\right]$$

$$= f(x)\left(-\frac{1}{x} - \frac{\log x - \mu}{\sigma^2 x}\right)^2 + f(x)\left[\frac{1}{x^2}\left(1 - \frac{1}{\sigma^2}\right) + \frac{\log x - \mu}{\sigma^2 x^2}\right]$$

$$= f(x)\frac{1}{x^2}\left[\left(1 + \frac{\log x - \mu}{\sigma^2}\right)^2 + \left(1 - \frac{1}{\sigma^2}\right) + \frac{\log x - \mu}{\sigma^2}\right].$$

We find inflection points by solving $f''(x) = 0$. Since $f(x) > 0$, this simplifies to:

$$\left(1 + \frac{\log x - \mu}{\sigma^2}\right)^2 + \left(1 - \frac{1}{\sigma^2}\right) + \frac{\log x - \mu}{\sigma^2} = 0$$

Let $z = \log x$. Substituting $z$, we get:

$$\left(1 + \frac{z - \mu}{\sigma^2}\right)^2 + \left(1 - \frac{1}{\sigma^2}\right) + \frac{z - \mu}{\sigma^2} = 0$$

$$1 + 2\frac{z - \mu}{\sigma^2} + \frac{(z - \mu)^2}{\sigma^4} + \left(1 - \frac{1}{\sigma^2}\right) + \frac{z - \mu}{\sigma^2} = 0$$

$$\frac{(z - \mu)^2}{\sigma^4} + \frac{3(z - \mu)}{\sigma^2} + \left(2 - \frac{1}{\sigma^2}\right) = 0$$

This is a quadratic equation in $z - \mu$. Let $a = \frac{1}{\sigma^4}$, $b = \frac{3}{\sigma^2}$, and $c = 2 - \frac{1}{\sigma^2}$. The equation becomes $a(z - \mu)^2 + b(z - \mu) + c = 0$, which can be solved using the quadratic formula, which gives us:

$$x = \exp\left(\mu + \frac{-b \pm \sqrt{b^2 - 4ac}}{2a}\right), \quad a = \frac{1}{\sigma^4}, \ b = \frac{3}{\sigma^2}, \ c = 2 - \frac{1}{\sigma^2}.$$

Plugging values back in, we get: $\exp(\mu + \frac{\sigma^2}{2}\left(-3 \pm \sqrt{1 + \frac{4}{\sigma^2}}\right))$.

### B.4 WEIBULL

The Weibull PDF is given by $f(x; k, \lambda) = \frac{k}{\lambda} \left(\frac{x}{\lambda}\right)^{k-1} e^{-\left(\frac{x}{\lambda}\right)^k}$; $k, \lambda > 0$, where $k$ is called shape and $\lambda$ is scale parameter. The first derivative of $f(x)$ is:

$$
\begin{aligned}
f'(x) &= \frac{d}{dx} \left[\frac{k}{\lambda} \left(\frac{x}{\lambda}\right)^{k-1} e^{-\left(\frac{x}{\lambda}\right)^k}\right] \\
&= \frac{k}{\lambda} \left[\frac{d}{dx} \left(\frac{x}{\lambda}\right)^{k-1} e^{-\left(\frac{x}{\lambda}\right)^k} + \left(\frac{x}{\lambda}\right)^{k-1} \frac{d}{dx} \left(e^{-\left(\frac{x}{\lambda}\right)^k}\right)\right] \\
&= \frac{k}{\lambda} \left[(k-1) \left(\frac{x}{\lambda}\right)^{k-2} \frac{1}{\lambda} e^{-\left(\frac{x}{\lambda}\right)^k} - k \left(\frac{x}{\lambda}\right)^{k-1} \frac{1}{\lambda} e^{-\left(\frac{x}{\lambda}\right)^k}\right] \\
&= f(x) \left[\frac{k-1}{x} - \frac{k}{\lambda} \left(\frac{x}{\lambda}\right)^{k-1}\right].
\end{aligned}
$$

The second derivative of $f(x)$ is:

$$
\begin{aligned}
f''(x) &= \frac{d}{dx} \left[f(x) \left(\frac{k-1}{x} - \frac{k}{\lambda} \left(\frac{x}{\lambda}\right)^{k-1}\right)\right] \\
&= f'(x) \left(\frac{k-1}{x} - \frac{k}{\lambda} \left(\frac{x}{\lambda}\right)^{k-1}\right) + f(x) \frac{d}{dx} \left(\frac{k-1}{x} - \frac{k}{\lambda} \left(\frac{x}{\lambda}\right)^{k-1}\right) \\
&= f'(x) \left(\frac{k-1}{x} - \frac{k}{\lambda} \left(\frac{x}{\lambda}\right)^{k-1}\right) + f(x) \left[-\frac{k-1}{x^2} - \frac{k(k-1)}{\lambda^2} \left(\frac{x}{\lambda}\right)^{k-2}\right] \\
&= f'(x) \left(\frac{k-1}{x} - \frac{k}{\lambda} \left(\frac{x}{\lambda}\right)^{k-1}\right) + f(x) \left[-\frac{k-1}{x^2} - \frac{k(k-1)}{\lambda^2} \left(\frac{x}{\lambda}\right)^{k-2}\right].
\end{aligned}
$$

To find the inflection points, we solve $f''(x) = 0$. Since $f(x) > 0$, we simplify to:

$$
\left(\frac{k-1}{x} - \frac{k}{\lambda} \left(\frac{x}{\lambda}\right)^{k-1}\right)^2 = -\frac{k-1}{x^2} - \frac{k(k-1)}{\lambda^2} \left(\frac{x}{\lambda}\right)^{k-2}
$$

$$
\left|\frac{k-1}{x} - \frac{k}{\lambda} \left(\frac{x}{\lambda}\right)^{k-1}\right| = \sqrt{\frac{k-1}{x^2} + \frac{k(k-1)}{\lambda^2} \left(\frac{x}{\lambda}\right)^{k-2}}.
$$

This gives two cases to solve:

$$
\frac{k-1}{x} - \frac{k}{\lambda} \left(\frac{x}{\lambda}\right)^{k-1} = \pm\sqrt{\frac{k-1}{x^2} + \frac{k(k-1)}{\lambda^2} \left(\frac{x}{\lambda}\right)^{k-2}}.
$$

Let us substitute $y = \frac{x}{\lambda}$, then we have:

$$
\frac{k-1}{\lambda y} - \frac{k}{\lambda} y^{k-1} = \pm\sqrt{\frac{k-1}{\lambda^2 y^2} + \frac{k(k-1)}{\lambda^2} y^{k-2}}
$$

$$
\frac{k-1}{y} - k y^{k-1} = \pm\sqrt{\frac{k-1}{y^2} + k(k-1) y^{k-2}}
$$

$$
\left(\frac{k-1}{y} - k y^{k-1}\right)^2 = \frac{k-1}{y^2} + k(k-1) y^{k-2}
$$

$$
\frac{(k-1)^2}{y^2} - 2k(k-1) y^{k-2} + k^2 y^{2k-2} = \frac{k-1}{y^2} + k(k-1) y^{k-2}
$$

We rearrange to get:

$$
\frac{(k-1)^2 - (k-1)}{y^2} - 2k(k-1) y^{k-2} - k(k-1) y^{k-2} + k^2 y^{2k-2} = 0
$$

$$
(k-1)(k-2) - 3k(k-1) y^k + k^2 y^{2k} = 0
$$

This is a quadratic equation in $z = y^k$, which gives:

$$k^2 z^2 - 3k(k-1)z + (k-1)(k-2) = 0,$$

then, using the quadratic formula:

$$z = \frac{3k(k-1) \pm \sqrt{9k^2(k-1)^2 - 4k^2(k-1)(k-2)}}{2k^2}.$$

We can simplify further:

$$9k^2(k-1)^2 - 4k^2(k-1)(k-2) = k^2(k-1)(9(k-1) - 4(k-2)) = k^2(k-1)(5k-1),$$

when we substitute back into the quadratic solution we get:

$$z = \frac{3k(k-1) \pm k\sqrt{(k-1)(5k-1)}}{2k^2} = \frac{3(k-1) \pm \sqrt{(k-1)(5k-1)}}{2k}.$$

In the original variables (from $z = y^k$ and $y = \frac{x}{\lambda}$) this gives us:

$$\left(\frac{x}{\lambda}\right)^k = \frac{3(k-1) \pm \sqrt{(k-1)(5k-1)}}{2k}$$

$$\frac{x}{\lambda} = \left(\frac{3(k-1) \pm \sqrt{(k-1)(5k-1)}}{2k}\right)^{1/k}$$

$$x = \lambda \left(\frac{1}{2}\right)^{1/k} \left(\frac{3(k-1) \pm \sqrt{(k-1)(5k-1)}}{k}\right)^{1/k}.$$

## C  DATA

### C.1  BECNHMARK DATA

Table 4 shows the summary statistics for real-world data used in experiments. Figure 6 shows the distribution of inter-event times, and Figure 7 shows sample sequences.

Table 4: Facts about the real-world data used in experiments.

| Dataset | Mark dim. | Majority mark | Number of sequences Train | Val | Test | Sequence lengths: "min–max (median)" Train | Val | Test |
|---|---|---|---|---|---|---|---|---|
| Amazon | 16 | 29.1% | 6454 | 922 | 1851 | 14 - 94 (42) | 15 - 94 (42) | 14 - 94 (42) |
| EQ | 7 | 43.7% | 3000 | 400 | 900 | 15 - 18 (16) | 15 - 18 (17) | 0 - 18 (16) |
| Reddit | 985 | 10.8% | 6000 | 2000 | 2000 | 28 - 100 (44) | 29 - 100 (45) | 29 - 100 (45) |
| Retweet | 3 | 49.4% | 20000 | 2000 | 2000 | 50 - 264 (90) | 50 - 264 (89) | 50 - 264 (90) |
| SO | 22 | 44.1% | 1401 | 401 | 401 | 41 - 101 (57) | 41 - 101 (58) | 41 - 101 (61) |
| Taobao | 17 | 44.3% | 1300 | 200 | 500 | 28 - 64 (61) | 31 - 64 (61) | 32 - 64 (61) |
| Taxi | 10 | 44.6% | 1400 | 200 | 400 | 36 - 38 (38) | 36 - 38 (38) | 36 - 38 (38) |

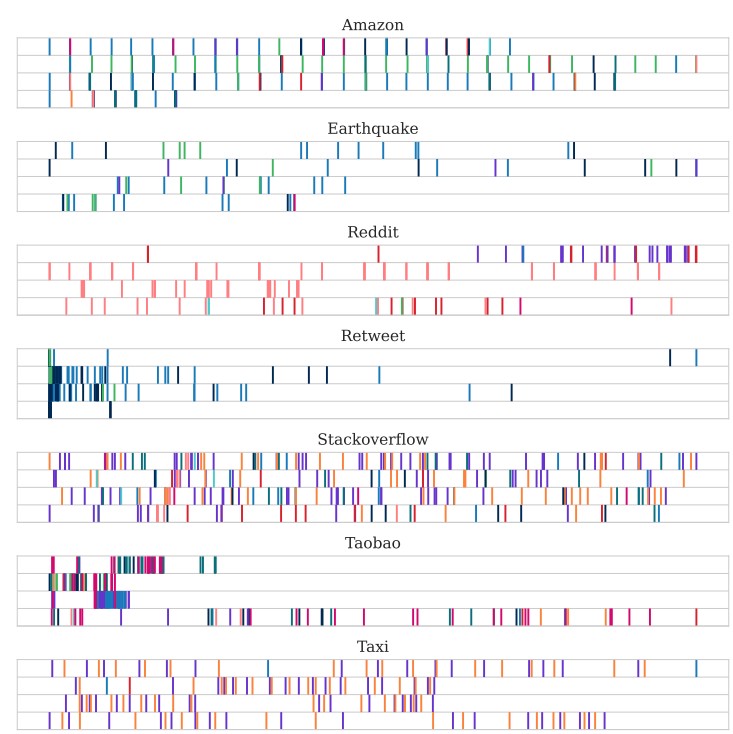

Figure 7: Four sequence examples for each benchmark dataset. Vertical lines correspond to the events, the space between the lines represents the time difference between events, different colors indicate different mark types.

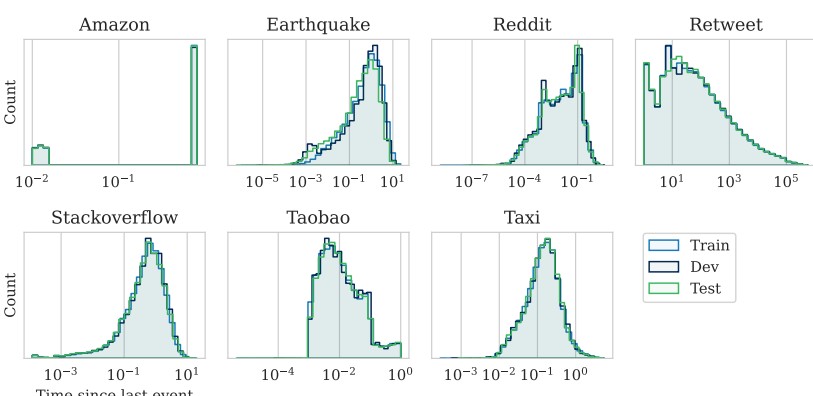

Figure 6: Histogram of times between consecutive events. Different colors are different data splits.

## C.2 LIMIT ORDER BOOK DATA

Table 5 describes the data fields in limit order book message data. We only keep submission (1) and deletion (3) types since they account for 94% of messages. Combining these two types with two directions gives us 4 different marks. Other possibilities, such as including size or price are possible but we leave this for future work.

| Field | Description |
|-------|-------------|
| Time | Seconds after midnight with decimal precision of at least milliseconds |
| Type | Categorical variable with the following possible values: |
| | 1. Submission of a new limit order |
| | 2. Cancellation (partial deletion of a limit order) |
| | 3. Deletion (total deletion of a limit order) |
| | 4. Execution of a visible limit order |
| | 5. Execution of a hidden limit order |
| | 6. Trading halt indicator |
| Order ID | Unique order reference number (assigned in order flow) |
| Size | Number of shares |
| Price | Dollar price |
| Direction | -1 for sell limit order, and 1 for buy limit order |

Table 5: Description of LOB data fields.

# D EXPERIMENTS

## D.1 HARDWARE

All experiments were conducted on a server equipped with a 40 core CPU at 2.40GHz and 754GB of RAM. The system features dual NVIDIA Tesla V100 GPUs, each with 16GB of dedicated VRAM.

## D.2 JUMP PROCESS SYNTHETIC DATA RESULTS

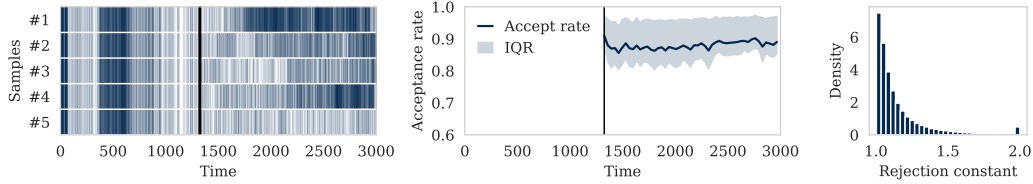

Figure 8: Qualitative results on jump process data. (Left) Five samples generated from the same initial sequence. Vertical blue lines indicate events, which results in visually darker areas for regimes with higher intensity. (Middle) Acceptance ratio over time. (Right) Rejection constant distribution.

## D.3 STATIONARITY OF REAL-WORLD DATA

We use an intensity-free model with a log-normal mixture distribution (Shchur et al., 2020). The full history model employs a GRU that updates its state with each event, while the Markov model processes events without retaining history. The no-history (renewal) model learns a single shared distribution for all events. Our model variations are implemented within the framework of Xue et al. (2024). In Table 6 we do not report Retweet RMSE results because of numerical instability which occurs computing this metric for all models.

## D.4 MONTE CARLO REJECTION CONSTANT

We compute the rejection constant necessary for exact baseline speculative sampling through a comprehensive numerical approach. Given a target distribution $f_T(t)$ and a proposal distribution $f_P(t)$, the method constructs a dense grid of evaluation points spanning the support of both distributions. At each point in this grid, we compute the ratio $f_T(t)/f_P(t)$. The rejection constant $M$ is then determined as the maximum value of this ratio across all evaluation points, effectively finding $M = \max_t \frac{f_T(t)}{f_P(t)}$. With enough points, this approach guarantees that $Mf_P(t) \geq f_T(t)$ for all $t$ in the domain, ensuring the correctness of the rejection sampling procedure. While computationally

| Metric | Data | TPP model type based on history | | |
|---|---|---|---|---|
| | | No history | Markov | Full history |
| Log-likelihood | Amazon | 0.205±0.035 | 0.231±0.040 | **0.513±0.026** |
| | Earthquake | -2.307±0.002 | -2.306±0.001 | **-2.017±0.012** |
| | Retweet | -9.892±0.000 | -9.892±0.000 | **-9.849±0.009** |
| | Stackoverflow | -2.689±0.001 | -2.679±0.002 | **-2.229±0.001** |
| | Taobao | 0.429±0.010 | 0.406±0.000 | **0.790±0.017** |
| | Taxi | -0.608±0.005 | -0.597±0.001 | **0.384±0.002** |
| Accuracy | Amazon | 0.290±0 (0.29) | 0.290±0 | **0.347±0.001** |
| | Earthquake | 0.451±0 (0.44) | 0.451±0 | **0.470±0.001** |
| | Retweet | 0.496±0 (0.49) | 0.496±0 | **0.601±0.004** |
| | Stackoverflow | 0.437±0 (0.44) | 0.437±0 | **0.473±0.000** |
| | Taobao | 0.422±0 (0.44) | 0.422±0 | **0.569±0.004** |
| | Taxi | 0.439±0 (0.45) | 0.439±0 | **0.907±0.001** |
| RMSE | Amazon | 3.867±5.035 | 0.486±0.001 | **0.462±0.000** |
| | Earthquake | **2.404±0.033** | 3.221±1.561 | 2.729±0.563 |
| | Stackoverflow | 1.549±0.041 | **1.529±0.015** | 2.233±0.654 |
| | Taobao | 3.641±5.295 | **0.166±0.005** | 6.859±2.314 |
| | Taxi | 13.573±12.906 | **0.437±0.015** | 0.449±0.041 |

Table 6: Model test log-likelihood, mark accuracy and time RMSE. Brackets in "No history" model's accuracy column indicate the majority class. Full history model is the best overall, indicating the processes are non-stationary.

| Exact | Data | MMD | | KL | | LLR | | Step |
|---|---|---|---|---|---|---|---|---|
| | | Autoreg | MC | Autoreg | MC | Autoreg | MC | |
| True | Amazon | 0.2 | 0.2 | 7.21 | 7.17 | -0.01 | -0.0 | 1.2608 |
| | EQ | 0.19 | 0.19 | 3.99 | 3.93 | -0.03 | -0.09 | 2.0946 |
| | Reddit | 0.19 | 0.2 | 6.11 | 6.28 | 0.03 | -0.08 | 1.6671 |
| | Retweet | 0.2 | 0.19 | 1.89 | 1.82 | -0.02 | -0.01 | 2.0967 |
| | SO | 0.19 | 0.19 | 6.91 | 6.98 | 0.02 | -0.0 | 1.5393 |
| | Taobao | 0.2 | 0.2 | 8.43 | 8.51 | -0.01 | -0.13 | 1.4982 |
| | Taxi | 0.19 | 0.19 | 2.78 | 2.78 | -0.0 | -0.0 | 1.0001 |
| False | Amazon | 0.2 | 0.2 | 7.2 | 7.23 | 0.01 | 0.01 | 2.5115 |
| | EQ | 0.19 | 0.2 | 3.84 | 3.98 | 0.0 | -0.1 | 2.9172 |
| | Reddit | 0.19 | 0.2 | 6.43 | 6.55 | 0.02 | -0.15 | 2.1583 |
| | Retweet | 0.19 | 0.2 | 1.89 | 1.88 | 0.0 | -0.03 | 2.4527 |
| | SO | 0.18 | 0.18 | 6.93 | 7.02 | 0.04 | -0.01 | 2.1158 |
| | Taobao | 0.2 | 0.2 | 8.38 | 8.63 | -0.0 | -0.16 | 1.7809 |
| | Taxi | 0.19 | 0.19 | 2.68 | 2.71 | 0.02 | 0.02 | 1.0004 |

Table 7: Speculative sampling using Monte Carlo (MC) approximation of the rejection constant, compared to the traditional one-by-one sampling (Autoreg). Speculative sampling is either exact (with tight bounds) or inexact (with loose bounds on categorical rejection constant and percentile computation). Inexact setting does not apply to autoregressive baseline. The reason for different values in "Autoreg" column for the same data is due to different random seeds.

intensive compared to our proposed method, the MC brute force method provides a reliable measure of the *optimal* rejection constant which can then be used to examine the gap in our approximation.

Table 7 shows the metrics of distance between the true autoregressive samples and speculative samples generated using the Monte Carlo rejection constant. For autoregressive column values, the distance is computed on two disjoint sets of samples from the same sampling procedure. This demonstrates the empirical distance between the values from the same distribution as a lower bound. It additionally shows the average achieved speculative step. We distinguish between exact and inexact approach where the latter has a looser definition of the bounds on which the grid is constructed and the categorical distribution rejection constant is computed with $\delta = 0.05$.

## D.5 METRICS

To quantify the statistical similarity between sequences generated by our speculative sampling method and conventional autoregressive sampling, we employ multiple complementary metrics.

**KL divergence per event.** We measure the Kullback-Leibler divergence between the empirical mark distributions of generated samples on an event-by-event basis. For discrete mark sequences $\mathbf{x}_p$ and $\mathbf{x}_q$ from distributions $P$ and $Q$ respectively, we compute:

$$\text{KL}_{\text{per event}} = \frac{1}{BL} \sum_{b=1}^{B} \sum_{l=1}^{L} \sum_{d=1}^{D} \hat{p}_{b,l}^{(d)} \log \frac{\hat{p}_{b,l}^{(d)}}{\hat{q}_{b,l}^{(d)}} \tag{14}$$

where $B$ is the batch size, $L$ is the sequence length, $D$ is the mark space dimension, and $\hat{p}_{b,l}^{(d)}$, $\hat{q}_{b,l}^{(d)}$ represent the empirical probability mass at dimension $d$ for event $l$ in batch $b$, computed using frequency counts across samples. A value closer to zero indicates greater similarity between distributions.

**Maximum Mean Discrepancy (MMD).** To compare the temporal aspects of generated sequences, we employ MMD with a Gaussian kernel to measure the distance between distributions of inter-arrival times. For each event position, we compute:

$$\text{MMD}_{b,l}(\mathbf{t}_p, \mathbf{t}_q) = \mathbb{E}[k(t_{p,b,l}, t'_{p,b,l})] + \mathbb{E}[k(t_{q,b,l}, t'_{q,b,l})] - 2\mathbb{E}[k(t_{p,b,l}, t_{q,b,l})] \tag{15}$$

where $k(\cdot, \cdot)$ is a Gaussian kernel with bandwidth selected via median L1 distance heuristic. The final MMD is averaged across all batch elements and event positions. MMD approaches zero as distributions become identical.

**Log-Likelihood Ratio.** This metric directly evaluates distributional agreement by comparing the log-probabilities assigned by the model to samples from different methods:

$$\text{LLR} = \frac{1}{BSL} \sum_{b=1}^{B} \sum_{s=1}^{S} \sum_{l=1}^{L} \left( \log p_{\text{new}}(t_{s,l}, x_{s,l}) - \log p_{\text{old}}(t_{s,l}, x_{s,l}) \right) \tag{16}$$

where $S$ is the number of samples per sequence, and the log-probabilities are computed using the same trained model. Values near zero indicate agreement between the sampling methods' output distributions.

For all metrics, we compute baseline comparisons by splitting samples from the conventional method into two halves and measuring the same statistics between them, providing a reference for expected variation within a single sampling method.

## D.6 ADDITIONAL RESULTS

Table 8 shows all the results for GRU encoder with hidden dimension of 256 and a log-normal mixture with 32 components. We test out different top-k values while measuring the divergence from the true samples, indicated by "Baseline". The speculative step is adjusted for different k, for 2 it becomes 10 and for 3 it is 15. As we can see, for most of the datasets larger top-k does not change the sample quality. The only dataset for which this is not the case is Taxi, which is a known issue discussed in the main text. Time constant and mark constant indicate the average rejection constant for the respective time and mark distributions.

Table 10 similarly shows results for a two layer transformer network, with a similar setup as for GRU. We sample 10 samples, each with 20 sequences and use a fixed speculative step of 5 for all k. Table 9 shows timing results for convolutional neural network encoder. We also show results for GRU encoder with different decoders, exponential in Table 11 and Weibull distribution in Table 12.

Measuring wall-clock time for rejection sampling reveals that depending on the speculative size and the way we construct a grid, the total time spent on computing the constant can be up to 100ms. This is the worst case for non-optimized code, and we expect that the speed can be improved significantly. The algorithm has linear complexity in the number of grid points, or log-linear if we need to sort them. The reason encoder is fast because (1) we use small models and (2) it is implemented using highly optimized native functions. We expect that using larger models on longer sequences combined with a faster optimization of rejection step, instead of the current didactic approach to code, will show that even for small acceptance rate we match or outperform the conventional approach.

| | Data Top-k | Amazon | EQ | Reddit | Retweet | SO | Taobao | Taxi |
|---|---|---|---|---|---|---|---|---|
| LLR | 1 | -0.02±2.07 | -0.13±2.11 | -0.15±3.57 | 0.02±1.42 | -0.01±2.47 | -0.07±2.74 | 0.02±1.41 |
| | 2 | -0.03±2.06 | -0.05±2.09 | -0.16±3.55 | 0.01±1.4 | -0.06±2.53 | -0.21±2.73 | 2.68±2.63 |
| | 3 | -0.05±2.07 | 0.03±2.06 | -0.21±3.48 | -0.03±1.4 | -0.07±2.56 | -0.24±2.71 | 2.81±2.59 |
| Baseline | | 0.02±2.06 | 0.04±2.13 | -0.06±3.56 | 0.02±1.46 | 0.01±2.43 | 0.02±2.73 | 0.02±1.41 |
| MMD | 1 | 0.2±0.2 | 0.2±0.17 | 0.19±0.16 | 0.19±0.14 | 0.18±0.16 | 0.2±0.15 | 0.19±0.16 |
| | 2 | 0.19±0.19 | 0.19±0.16 | 0.19±0.16 | 0.19±0.14 | 0.2±0.15 | 0.2±0.15 | 0.21±0.17 |
| | 3 | 0.2±0.2 | 0.18±0.16 | 0.2±0.17 | 0.19±0.14 | 0.18±0.16 | 0.2±0.15 | 0.21±0.17 |
| Baseline | | 0.2±0.19 | 0.19±0.16 | 0.19±0.17 | 0.2±0.14 | 0.19±0.16 | 0.2±0.14 | 0.19±0.15 |
| KL per event | 1 | 7.33±4.17 | 3.98±3.85 | 6.4±6.34 | 1.92±3.0 | 6.97±4.59 | 8.59±4.53 | 2.71±3.0 |
| | 2 | 7.26±4.18 | 3.99±3.81 | 6.45±6.36 | 1.87±2.93 | 6.93±4.56 | 8.34±4.58 | 6.38±6.08 |
| | 3 | 7.28±4.21 | 3.86±3.79 | 6.26±6.32 | 1.99±3.03 | 7.0±4.6 | 8.63±4.63 | 6.27±6.41 |
| Baseline | | 7.26±4.16 | 3.9±3.81 | 6.36±6.26 | 1.89±2.95 | 6.93±4.55 | 8.33±4.43 | 2.64±2.94 |
| Rank correlation | 1 | 0.0±0.05 | 0.01±0.11 | 0.0±0.06 | 0.0±0.05 | 0.0±0.05 | -0.0±0.05 | -0.0±0.06 |
| | 2 | 0.0±0.05 | 0.0±0.1 | 0.01±0.06 | 0.0±0.06 | -0.02±0.06 | -0.0±0.05 | 0.01±0.05 |
| | 3 | -0.0±0.04 | -0.02±0.11 | 0.02±0.06 | 0.01±0.06 | -0.0±0.05 | 0.0±0.05 | 0.0±0.04 |
| Baseline | | 0.0±0.05 | -0.01±0.11 | -0.0±0.06 | -0.0±0.06 | -0.01±0.05 | -0.0±0.05 | 0.0±0.05 |
| Time constant | 1 | 1.0656 | 1.5293 | 1.8412 | 1.0 | 1.3579 | 1.4391 | 3.5415 |
| | 2 | 1.0711 | 1.6574 | 1.8747 | 1.0 | 1.3891 | 1.4742 | 2.6866 |
| | 3 | 1.0785 | 1.8223 | 1.8644 | 1.0 | 1.4136 | 1.4952 | 2.5601 |
| Mark constant | 1 | 1.4906 | 1.2353 | 2.3063 | 1.2389 | 10.7231 | 4.2384 | 4228.5493 |
| | 2 | 1.5876 | 1.2418 | 2.4749 | 1.2579 | 19.9326 | 4.312 | 509.4294 |
| | 3 | 1.6273 | 1.2454 | 2.6028 | 1.2751 | 48.905 | 4.3765 | 498.4741 |
| Encoder runtime | 1 | 16.66±1.16 | 20.61±6.32 | 25.05±7.17 | 13.81±1.14 | 29.85±5.45 | 37.56±2.88 | 54.33±1.12 |
| | 2 | 11.4±0.98 | 10.99±2.41 | 19.2±5.24 | 7.73±0.75 | 17.36±3.03 | 22.12±1.66 | 29.86±1.61 |
| | 3 | 7.47±0.73 | 7.82±1.24 | 13.81±3.72 | 5.94±0.55 | 13.13±2.18 | 16.22±1.39 | 22.62±1.57 |
| Baseline | | 36.36±0.73 | 46.53±12.39 | 37.89±2.31 | 39.24±2.27 | 49.02±0.62 | 49.06±0.91 | 49.18±1.55 |
| Decoder runtime | 1 | 37.04±2.6 | 45.83±14.22 | 55.34±16.26 | 29.31±2.22 | 59.83±11.42 | 76.34±6.05 | 111.12±2.29 |
| | 2 | 23.11±2.07 | 21.32±4.31 | 36.2±10.33 | 14.11±1.19 | 30.87±5.93 | 40.45±3.58 | 55.5±4.04 |
| | 3 | 12.87±1.33 | 13.4±1.77 | 22.94±6.61 | 9.4±0.84 | 21.46±4.07 | 27.12±2.55 | 38.98±3.47 |
| Baseline | | 42.04±0.63 | 54.09±14.5 | 43.41±2.32 | 44.13±2.01 | 50.61±0.66 | 50.71±1.24 | 51.13±2.14 |
| Sample runtime | 1 | 12.81±0.89 | 16.11±5.08 | 19.22±5.66 | 10.27±0.83 | 22.35±4.26 | 28.12±2.49 | 40.71±1.04 |
| | 2 | 8.15±0.72 | 7.78±1.66 | 13.54±3.88 | 5.1±0.5 | 12.3±2.36 | 15.71±1.34 | 21.7±1.54 |
| | 3 | 4.74±0.49 | 5.09±0.7 | 9.1±2.62 | 3.53±0.35 | 8.88±1.74 | 10.99±1.19 | 15.61±1.28 |
| Baseline | | 47.84±0.83 | 62.69±18.08 | 49.53±2.84 | 50.35±2.6 | 60.72±0.96 | 60.79±1.16 | 61.03±2.29 |
| Step | 1 | 2.8905 | 3.0387 | 2.1848 | 3.7707 | 2.1695 | 1.7783 | 1.0003 |
| | 2 | 5.9309 | 6.1882 | 4.3222 | 8.2337 | 4.282 | 3.4225 | 2.1283 |
| | 3 | 8.8409 | 9.1424 | 6.5014 | 12.6512 | 6.225 | 5.0025 | 3.0359 |

Table 8: Results for GRU encoder and log-normal mixture.

| | Top-k | Amazon | EQ | Reddit | Retweet | SO | Taobao | Taxi |
|---|---|---|---|---|---|---|---|---|
| LLR | 1 | 4.18±5.31 | 7.81±4.72 | N/A | 4.71±4.88 | 5.25±4.8 | 1.07±5.0 | 2.16±3.98 |
| | 2 | 9.02±4.59 | 9.88±3.9 | 9.09±6.05 | 5.76±5.1 | 8.0±4.29 | 4.81±5.07 | 3.07±4.01 |
| | 3 | 8.84±4.65 | 9.96±3.96 | 9.32±6.03 | 4.96±5.06 | 8.08±4.16 | 6.59±4.99 | 3.21±3.97 |
| Baseline | | 0.01±3.57 | 0.02±2.5 | -0.03±6.91 | -0.0±3.15 | -0.0±3.36 | -0.04±5.3 | -0.0±3.59 |
| MMD | 1 | 0.36±0.21 | 0.19±0.16 | 0.47±0.26 | 0.75±0.34 | 0.72±0.34 | 0.86±0.23 | 1.02±0.24 |
| | 2 | 0.33±0.21 | 0.19±0.16 | 0.38±0.25 | 0.36±0.21 | 0.22±0.16 | 0.33±0.23 | 0.9±0.3 |
| | 3 | 0.35±0.22 | 0.19±0.16 | 0.38±0.27 | 0.42±0.23 | 0.21±0.15 | 0.39±0.33 | 0.78±0.32 |
| Baseline | | 0.19±0.14 | 0.19±0.16 | 0.2±0.14 | 0.19±0.14 | 0.19±0.13 | 0.18±0.14 | 0.18±0.14 |
| KL per event | 1 | 11.14±3.76 | 5.28±4.47 | 15.94±1.88 | 1.14±2.73 | 13.18±3.24 | 13.0±3.43 | 8.25±4.54 |
| | 2 | 11.19±3.76 | 4.43±4.01 | 15.71±2.11 | 1.31±3.17 | 12.94±3.33 | 12.0±3.62 | 8.09±4.37 |
| | 3 | 11.24±3.78 | 4.16±3.89 | 15.77±1.99 | 1.1±2.64 | 13.01±3.31 | 11.77±3.65 | 7.58±4.39 |
| Baseline | | 11.11±3.78 | 4.22±3.93 | 15.24±2.38 | 1.17±2.59 | 13.01±3.24 | 11.58±3.69 | 6.33±4.11 |
| Rank correlation | 1 | 0.01±0.05 | -0.0±0.05 | -0.01±0.05 | 0.0±0.03 | -0.0±0.03 | 0.01±0.05 | -0.0±0.05 |
| | 2 | -0.01±0.04 | 0.01±0.05 | 0.0±0.05 | 0.0±0.03 | 0.0±0.04 | -0.0±0.05 | -0.0±0.05 |
| | 3 | 0.01±0.05 | -0.0±0.05 | 0.01±0.05 | 0.0±0.04 | 0.01±0.05 | 0.0±0.05 | -0.0±0.06 |
| Baseline | | -0.0±0.04 | 0.0±0.04 | -0.01±0.04 | -0.01±0.02 | 0.0±0.03 | -0.0±0.04 | -0.0±0.05 |
| Time constant | 1 | 2682.1531 | 115.6433 | N/A | 756.444 | 76.4793 | 43.3235 | 298.3471 |
| | 2 | 1131.7275 | 167.2671 | N/A | 383.369 | 175.6843 | 221.8055 | 79.851 |
| | 3 | 1111.671 | 153.6114 | N/A | 238.7219 | 186.8324 | 362.6418 | 88.6586 |
| Mark constant | 1 | 33.2394 | 108.5231 | 983.8811 | 155.4943 | 63.4784 | 10.5921 | 6.6346 |
| | 2 | 33.736 | 84.2254 | 442.0354 | 107.2799 | 66.3883 | 27.6739 | 9.4718 |
| | 3 | 26.9968 | 64.4483 | 337.1147 | 89.9403 | 45.942 | 45.3992 | 12.4013 |
| Step | 1 | 1.3357 | 1.3856 | 1.3674 | 1.6934 | 1.292 | 1.1466 | 1.1433 |
| | 2 | 2.6618 | 2.6933 | 2.913 | 4.7831 | 2.6727 | 2.3286 | 2.2001 |
| | 3 | 4.0362 | 4.3148 | 4.6114 | 7.0428 | 4.0173 | 3.8323 | 3.2598 |

Table 9: Results for CNN encoder and log-normal mixture.

| | Top-k | Amazon | EQ | Reddit | Retweet | SO | Taobao | Taxi |
|---|---|---|---|---|---|---|---|---|
| LLR | 1 | 0.51±1.94 | -1.1±2.71 | 0.9±3.48 | -0.01±7.0 | 0.56±2.75 | 0.75±2.79 | 0.87±1.7 |
| | 2 | 0.44±2.01 | -0.54±2.63 | 0.65±3.49 | -1.43±6.1 | 0.39±2.72 | 0.38±2.89 | 0.86±1.65 |
| | 3 | 0.27±2.03 | -0.34±2.53 | 0.53±3.52 | -2.12±5.65 | 0.28±2.78 | 0.08±2.93 | 0.94±1.68 |
| Baseline | | 0.01±1.97 | -0.03±2.11 | 0.0±3.49 | 0.06±6.54 | 0.01±2.72 | -0.01±2.91 | -0.02±1.14 |
| MMD | 1 | 0.32±0.34 | 0.33±0.22 | 0.22±0.21 | 0.24±0.17 | 0.2±0.18 | 0.19±0.16 | 0.27±0.22 |
| | 2 | 0.31±0.34 | 0.28±0.2 | 0.21±0.19 | 0.3±0.2 | 0.19±0.17 | 0.2±0.15 | 0.26±0.21 |
| | 3 | 0.28±0.31 | 0.25±0.19 | 0.2±0.18 | 0.36±0.22 | 0.19±0.17 | 0.2±0.15 | 0.27±0.22 |
| Baseline | | 0.2±0.2 | 0.19±0.17 | 0.19±0.16 | 0.2±0.15 | 0.18±0.16 | 0.19±0.15 | 0.19±0.15 |
| KL per event | 1 | 10.32±4.86 | 4.45±4.29 | 5.56±6.59 | 1.74±3.4 | 6.9±4.89 | 7.82±5.55 | 5.46±6.64 |
| | 2 | 9.64±4.72 | 4.38±4.21 | 5.2±6.38 | 1.63±3.06 | 6.69±4.8 | 6.84±5.18 | 5.84±6.79 |
| | 3 | 9.16±4.82 | 4.61±4.35 | 4.83±6.04 | 1.52±2.84 | 6.71±4.81 | 6.41±5.15 | 5.58±6.66 |
| Baseline | | 6.95±4.38 | 3.81±3.86 | 3.55±5.02 | 1.51±2.77 | 5.65±4.37 | 4.52±4.02 | 1.14±2.08 |
| Rank correlation | 1 | 0.0±0.06 | -0.0±0.08 | -0.01±0.06 | -0.01±0.09 | -0.0±0.06 | 0.0±0.06 | 0.0±0.06 |
| | 2 | 0.01±0.06 | -0.01±0.08 | 0.01±0.06 | -0.0±0.1 | 0.0±0.05 | 0.0±0.05 | -0.01±0.06 |
| | 3 | 0.0±0.06 | 0.01±0.07 | -0.0±0.05 | -0.01±0.1 | -0.0±0.06 | 0.0±0.06 | 0.0±0.05 |
| Baseline | | 0.0±0.06 | -0.0±0.06 | 0.0±0.05 | -0.02±0.08 | -0.0±0.04 | -0.0±0.06 | -0.0±0.06 |
| Time constant | 1 | 3.0293 | 2.9214 | 1.6733 | 1.6153 | 1.721 | 1.5309 | 5.1346 |
| | 2 | 3.0626 | 2.3598 | 1.5995 | 1.273 | 1.5164 | 1.4856 | 4.6703 |
| | 3 | 2.8964 | 2.1497 | 1.5658 | 1.1631 | 1.4477 | 1.4743 | 4.4259 |
| Mark constant | 1 | 12.9586 | 5.345 | 81.0723 | 2.5292 | 11.8475 | 13.0482 | 558.3685 |
| | 2 | 9.2925 | 4.4702 | 62.7646 | 2.1753 | 10.3675 | 11.5164 | 650.7604 |
| | 3 | 8.1383 | 3.8468 | 46.027 | 1.9539 | 8.7498 | 9.9784 | 697.7171 |
| Step | 1 | 2.1756 | 1.5597 | 2.9815 | 2.8391 | 1.8302 | 2.9812 | 1.1026 |
| | 2 | 4.2941 | 3.3363 | 6.0441 | 6.4851 | 3.749 | 5.4506 | 2.1793 |
| | 3 | 6.0612 | 5.2772 | 8.8392 | 10.4488 | 5.6055 | 7.4253 | 3.2466 |

Table 10: Results for transformer encoder and log-normal mixture.

| | Top-k | Amazon | EQ | Reddit | Retweet | SO | Taobao | Taxi |
|---|---|---|---|---|---|---|---|---|
| LLR | 1 | -0.02±1.79 | -0.09±1.82 | -0.11±2.56 | 0.0±0.07 | -0.01±2.47 | -0.17±1.55 | -0.04±1.32 |
| | 2 | -0.04±1.79 | -0.11±1.77 | -0.11±2.53 | 0.01±0.08 | -0.08±2.51 | -0.21±1.56 | 2.91±2.8 |
| | 3 | -0.05±1.79 | -0.15±1.71 | -0.14±2.54 | 0.02±0.08 | -0.12±2.54 | -0.25±1.56 | 2.93±2.82 |
| Baseline | | -0.0±1.79 | 0.01±1.79 | 0.05±2.54 | -0.0±0.07 | 0.01±2.44 | 0.01±1.54 | -0.03±1.32 |
| MMD | 1 | 0.19±0.16 | 0.2±0.17 | 0.19±0.16 | 0.18±0.16 | 0.18±0.16 | 0.19±0.16 | 0.19±0.16 |
| | 2 | 0.19±0.16 | 0.2±0.17 | 0.19±0.16 | 0.19±0.16 | 0.19±0.16 | 0.19±0.15 | 0.19±0.16 |
| | 3 | 0.19±0.15 | 0.2±0.18 | 0.19±0.16 | 0.19±0.16 | 0.19±0.16 | 0.19±0.16 | 0.19±0.16 |
| Baseline | | 0.19±0.16 | 0.2±0.17 | 0.19±0.16 | 0.19±0.16 | 0.19±0.16 | 0.19±0.15 | 0.19±0.16 |
| KL per event | 1 | 7.64±4.26 | 4.19±3.92 | 6.59±6.37 | 1.72±2.86 | 6.95±4.52 | 7.84±4.59 | 2.7±3.34 |
| | 2 | 7.53±4.29 | 4.18±3.95 | 6.5±6.38 | 1.78±2.95 | 7.08±4.55 | 7.85±4.4 | 6.92±5.83 |
| | 3 | 7.62±4.28 | 4.23±3.95 | 6.47±6.4 | 1.73±2.86 | 6.97±4.56 | 7.75±4.5 | 6.99±6.54 |
| Baseline | | 7.61±4.27 | 4.09±3.89 | 6.43±6.27 | 1.73±2.88 | 7.0±4.57 | 7.66±4.48 | 2.69±3.31 |
| Rank correlation | 1 | 0.0±0.05 | -0.01±0.16 | 0.01±0.06 | 0.0±0.01 | -0.0±0.06 | -0.0±0.06 | 0.01±0.08 |
| | 2 | 0.0±0.05 | -0.0±0.15 | 0.0±0.09 | -0.0±0.0 | -0.0±0.05 | -0.0±0.07 | -0.0±0.04 |
| | 3 | -0.0±0.05 | 0.01±0.13 | -0.01±0.08 | 0.0±0.0 | 0.01±0.06 | -0.01±0.06 | -0.0±0.06 |
| Baseline | | -0.0±0.05 | -0.0±0.14 | -0.01±0.08 | 0.0±0.01 | -0.0±0.06 | 0.0±0.07 | 0.0±0.09 |
| Time constant | 1 | 1.2667 | 1.9845 | 1.5389 | 1.0026 | 1.4444 | 1.4892 | 3.327 |
| | 2 | 1.2805 | 2.2042 | 1.706 | 1.0053 | 1.5484 | 1.5257 | 3.0401 |
| | 3 | 1.286 | 2.3504 | 1.7577 | 1.0068 | 1.6356 | 1.5492 | 2.7018 |
| Mark constant | 1 | 1.7289 | 1.3241 | 2.3114 | 1.1298 | 11.1859 | 4.3067 | 7021.4775 |
| | 2 | 1.8439 | 1.3375 | 2.507 | 1.1448 | 18.6992 | 4.3776 | 970.8654 |
| | 3 | 1.8874 | 1.3411 | 2.5984 | 1.1519 | 37.7343 | 4.4245 | 1134.9554 |
| Step | 1 | 2.2616 | 2.5223 | 2.5251 | 4.1626 | 2.3546 | 1.8214 | 1.0004 |
| | 2 | 4.4213 | 5.1332 | 5.1172 | 8.9544 | 4.5941 | 3.4815 | 2.1796 |
| | 3 | 6.462 | 7.6076 | 7.5137 | 13.8284 | 6.6238 | 5.1294 | 3.0276 |

Table 11: Results for GRU encoder and exponential distribution.

|  | Data Top-k | Amazon | EQ | Reddit | Retweet | SO | Taxi |
|---|---|---|---|---|---|---|---|
| LLR | 1 | -0.03±1.89 | -0.16±1.99 | -0.11±4.5 | -0.01±7.23 | -0.05±2.44 | -0.01±1.09 |
|  | 2 | -0.03±1.92 | -0.09±1.99 | -0.12±4.46 | -0.04±7.19 | -0.14±2.5 | 3.07±2.69 |
|  | 3 | -0.07±1.9 | -0.03±2.0 | -0.08±4.47 | -0.02±7.24 | -0.21±2.57 | 3.03±2.71 |
| Baseline |  | 0.0±1.92 | -0.01±1.97 | 0.04±4.47 | -0.03±7.24 | 0.01±2.4 | 0.01±1.11 |
| MMD | 1 | 0.19±0.16 | 0.2±0.16 | 0.2±0.15 | 0.2±0.14 | 0.19±0.16 | 0.18±0.15 |
|  | 2 | 0.18±0.16 | 0.2±0.16 | 0.2±0.15 | 0.2±0.14 | 0.19±0.16 | 0.2±0.16 |
|  | 3 | 0.19±0.15 | 0.19±0.16 | 0.2±0.15 | 0.19±0.13 | 0.19±0.16 | 0.2±0.16 |
| Baseline |  | 0.18±0.16 | 0.19±0.15 | 0.2±0.15 | 0.2±0.13 | 0.18±0.16 | 0.18±0.16 |
| KL per event | 1 | 7.46±4.22 | 4.14±3.89 | 7.12±6.66 | 1.28±2.52 | 7.04±4.54 | 2.51±3.05 |
|  | 2 | 7.52±4.22 | 4.1±3.83 | 6.93±6.56 | 1.33±2.59 | 7.06±4.58 | 7.78±6.37 |
|  | 3 | 7.56±4.26 | 4.2±3.9 | 6.97±6.51 | 1.37±2.64 | 7.05±4.57 | 7.14±6.78 |
| Baseline |  | 7.46±4.25 | 4.12±3.87 | 6.89±6.51 | 1.34±2.58 | 6.99±4.51 | 2.53±3.05 |
| Rank correlation | 1 | 0.0±0.05 | 0.0±0.09 | -0.0±0.06 | 0.01±0.05 | -0.01±0.06 | 0.01±0.07 |
|  | 2 | 0.0±0.05 | -0.0±0.08 | 0.0±0.05 | 0.01±0.05 | -0.01±0.05 | -0.01±0.04 |
|  | 3 | -0.0±0.05 | 0.01±0.09 | 0.0±0.06 | 0.0±0.05 | -0.0±0.05 | 0.0±0.04 |
| Baseline |  | 0.01±0.05 | 0.01±0.08 | 0.01±0.05 | 0.01±0.05 | -0.01±0.05 | -0.01±0.06 |
| Time constant | 1 | 1.2976 | 1.3292 | 1.1461 | 1.0479 | 1.4029 | 10.4097 |
|  | 2 | 1.3104 | 1.3891 | 1.1776 | 1.0519 | 1.6359 | 9.5996 |
|  | 3 | 1.3221 | 1.4222 | 1.2016 | 1.0562 | 1.8774 | 9.7414 |
| Mark constant | 1 | 1.7017 | 1.2497 | 3.4656 | 1.1285 | 10.0589 | 9639.1475 |
|  | 2 | 1.8175 | 1.2626 | 3.8273 | 1.142 | 23.7208 | 886.7015 |
|  | 3 | 1.8802 | 1.263 | 3.97 | 1.1512 | 67.8345 | 1109.8258 |
| Step | 1 | 2.2871 | 3.1127 | 2.5807 | 3.9662 | 2.3344 | 1.0002 |
|  | 2 | 4.4654 | 6.4353 | 5.115 | 8.5965 | 4.5189 | 2.1213 |
|  | 3 | 6.486 | 9.5578 | 7.5811 | 13.1784 | 6.4567 | 3.0229 |

Table 12: Results for GRU encoder and Weibull distribution.

## D.7 LIMIT ORDER BOOKS

The figures present a comprehensive statistical comparison between conventional and speculative sampling methods for limit order book data. In Figure 9, transition matrices are constructed by calculating the probabilities of state-to-state transitions for both time deltas and mark types across three conditions: original data, conventional sampling, and speculative sampling. These matrices reveal that both sampling methods preserve the underlying transition dynamics of the data.

In the main text, Figure 4 displays empirical event sequences visualized as vertical lines on timelines (with colors representing different mark types), providing qualitative evidence that both sampling approaches generate visually similar patterns. Each sequence is prompted with the same initial sequence.

Figure 10 plots the accumulation of events by mark type over time, showing mean counts with standard deviation bands for both sampling methods across the four order types.

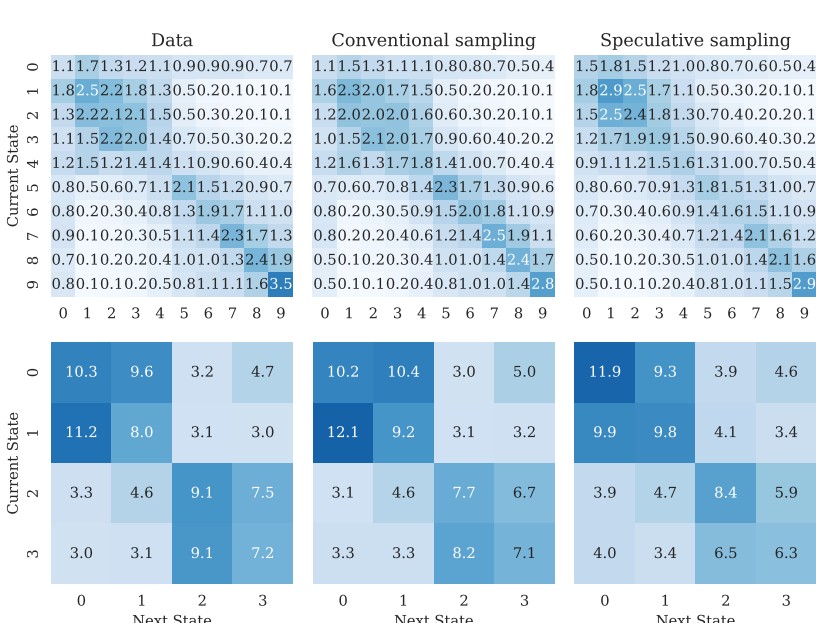

Figure 9: Limit order book transition matrix between two consecutive events. (Left) True transition matrix between the previous and next state. (Middle) Transition matrix obtained with conventional sampling. (Right) Transition matrix computed on samples coming from a speculative sampling scheme.

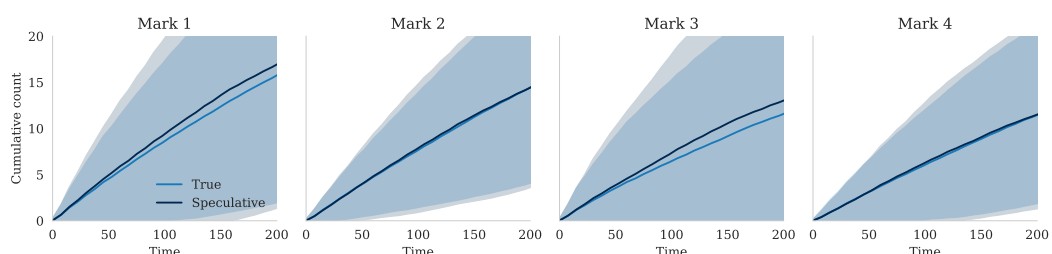

Figure 10: Limit order book cumulative event counts per mark dimension.