# OpenReview forum: "Speculative Sampling for Parametric Temporal Point Processes"
_ICLR.cc/2026/Conference — Submitted to ICLR 2026_

### Official Review · Reviewer_koMV · 2025-10-18

**Soundness:** 3
**Presentation:** 3
**Contribution:** 2
**Rating:** 4
**Confidence:** 3

**Summary:**

This paper proposes a novel algorithm based on rejection sampling that enables exact sampling of multiple future values from existing TPP models, in parallel, and without requiring any architectural changes or retraining. The authors provided theoretical guarantees, which makes the algorithm solid. Experiments demonstrate the good performance and potential application.

**Strengths:**

1. The paper is well-organized and the writing is clear.
2. The paper presents a parallel sampling method that accelerates TPP sampling without requiring architectural changes or retraining. It is a novel adaptation of the "speculative sampling" paradigm from language modeling to TPPs.
3. A key strength is the rigorous theoretical foundation, complete with derived rejection constants for distributions like Exponential, Gamma, Log-Normal, and Weibull, as well as a proven error bound that strengthens confidence in the method.

**Weaknesses:**

1. The algorithm's reliance on computing the rejection constant M via piecewise-linear approximation raises concerns. Key issues include whether the grid is sufficiently dense and if the approximation error for multi-modal distributions could lead to an underestimated M, thereby violating the guarantee of exact sampling.
2. Relying on the first rejected sample to terminate a speculative block can reduce parallelism, particularly for high-variability processes where early rejection is likely.

**Questions:**

1. The piecewise-linear approximation for estimating the rejection constant M in complex distributions raises several concerns. The tightness of this approximation critically depends on the selection and number of grid points, yet the paper lacks a principled method for this choice (e.g., based on density curvature). An overly loose bound (large M) inflates the rejection rate, hurting efficiency, while an overly tight one risks violating the exact sampling guarantee. It is also unclear if this method remains computationally feasible for highly complex or high-dimensional distributions.
2. A major limitation is the algorithm's dependence on the average accepted length for efficiency. For processes with strong temporal dependencies, the overhead from parallel verification and M computation may not be amortized, leading to performance worse than standard sampling. The paper would be strengthened by exploring more robust strategies for these cases beyond top-k, such as backtracking or partial re-sampling mechanisms.
3. Is the method particularly sensitive to the calibration quality of the underlying model? Have the authors investigated the correlation between model prediction uncertainty and inferred sampling efficiency?
4. Could the authors provide a detailed performance analysis showing what percentage of the total sampling time is spent computing the rejection constant in a typical experimental setup? Is there a critical point below which the total cost of the algorithm exceeds that of traditional autoregressive sampling?
5. Why not compare your approach to a model trained directly for multi-step forecasting? For example HYPRO [1] and Add-and-Thin [2]. Even if the latter has slightly lower sample quality, it can be sampled very quickly. This "quality-speed" trade-off is worth exploring.
6. For datasets with strong, near-deterministic transition rules like Taxi, is it possible to modify the proposal distribution to better capture this structure? Is there any empirical experiment results?

[1] HYPRO: A Hybridly Normalized Probabilistic Model for Long-Horizon Prediction of Event Sequences. NeurIPS 2022

[2] Add and thin: Diffusion for temporal point processes. NeurIPS 2023

---

> ### Author Response · Authors · 2025-11-14
>
> Thank you for your valuable feedback.
>
> **1. Concern about piecewise-linear bounds and multimodal distributions**
>
> The bounds we construct are not approximations--they are strict upper and lower bounds, ensuring that the rejection constant $M$ is never underestimated. This guarantees exact sampling as stated in Theorem 3.1.
>
> Our treatment of mixture distributions is rigorous and provides a strict bound on $M$.
>
> If two functions have bounding functions, the sum of the original functions is bounded by the sum of their bounding functions. We use piecewise linear functions to bound densities exactly.
>
> For mixtures, each component has its own grid points. It is straightforward to take the union of these grids and define piecewise linear bounds on the shared grid. Both components remain correctly bounded, and their sum is trivial to compute.
>
> We formalize this in Lemma A.2 (any density can be exactly bounded by piecewise linear functions) and Lemma A.3 (combining mixture components). Figure 2 illustrates this approach and confirms that multimodality does not pose an issue.
>
>
> **2. Impact of early rejection on parallelism**
>
> This is not an issue, even in batched sampling. In practice, early rejection does not degrade performance-our experiments confirm consistent runtime improvements. Even when one sequence rejects early, others continue independently, and we do not discard samples. Over long sequences, acceptance rates balance out, so batched sampling remains efficient.
>
>
> ## Questions
>
> **1. Piecewise-linear approximation and grid selection**
>
> As noted earlier, the grid construction for mixture distributions guarantees a correct bound and exact sampling. Each component uses an exact grid with valid bounds, and combining them is straightforward (Lemma A.3). While the tightness of the bound depends on the specific density and its parameters, our empirical results show this is not a practical issue. For example, we successfully used 32-component mixtures without difficulty, which is more than sufficient for modeling one-dimensional inter-event time densities. Even with high-dimensional mark distributions, the approach remained computationally feasible in all experiments.
>
>
> **2. Dependence on accepted length and alternative strategies**
>
> Thank you for this suggestion. Exploring strategies such as backtracking or partial re-sampling is an interesting direction for future work. Since our experiments show that top-k already performs robustly, we did not prioritize this in the current work.
>
>
> **3. Sensitivity to model calibration and uncertainty**
>
> Our Hawkes process experiments (Figure 3) illustrate how uncertainty affects efficiency. As interactions between marks become more deterministic, efficiency decreases. In the extreme case of fully deterministic transitions (Taxi dataset), speculative sampling cannot outperform traditional sampling. We argue that Taxi is not an ideal benchmark for mark modeling since predicting the next mark requires only the previous event. Hawkes provides a better controlled example: even under strong mark dependence, our method consistently improves efficiency. Real-world datasets further confirm this trend.
>
>
> **4. Performance breakdown for rejection constant computation**
>
> We will include detailed tables comparing rejection constant computation time to total sampling time. In our current implementation, rejection sampling was designed for clarity rather than speed, which makes this step slower than optimal. In the worst case, it accounts for up to 50% of total time. This means accepting at least three steps is necessary to outperform traditional sampling. We believe optimized implementations can significantly reduce this overhead.
>
>
> **5. Comparison to multi-step forecasting models**
>
> As noted in Section 4, our method accelerates sampling for existing autoregressive models without altering them. Comparing model accuracy to Add-and-Thin or HYPRO would not change this. Add-and-Thin reports slower sampling than autoregressive baselines, while HYPRO introduces an energy-based reweighting mechanism for long-horizon predictions, an orthogonal approach. In fact, our technique could be integrated into HYPRO to speed up its autoregressive component without affecting accuracy.
>
>
> **6. Handling near-deterministic transitions (Taxi dataset)**
>
> We experimented with using a mixture of two previous mark distributions instead of one, which significantly improved Taxi performance. Future work could adapt mixture weights dynamically or incorporate longer history. We omitted these refinements to keep the method simple and focused, but will mention them in the revised version.
>
>
> Thank you again for your thoughtful review and constructive feedback. We believe our clarifications address your concerns and strengthen the presentation of our work.

---

### Official Review · Reviewer_v7DL · 2025-10-31

**Soundness:** 3
**Presentation:** 3
**Contribution:** 3
**Rating:** 6
**Confidence:** 2

**Summary:**

This paper addresses the efficiency bottleneck of sampling from autoregressive temporal point processes (TPPs). The authors propose a novel speculative sampling algorithm based on rejection sampling that enables efficient generation of multiple future events. The approach can be applied to existing parametric TPP models without requiring any architectural changes or retraining. The core idea is to use a proposal distribution to generate several candidate events at once. These proposed events are then fed back into the model in a single parallel pass to compute their target (true) distribution and decide whether each sample is rejected or accepted. The method is supported by a principled theoretical foundation for computing the necessary rejection constant. Experiments on real-world datasets confirm that this approach provides substantial runtime speedups while maintaining the exact statistical properties of the original model.

**Strengths:**

1. The work is well-motivated, addressing the critical bottleneck of inefficient sampling in autoregressive Temporal Point Processes (TPPs). Addresses this major limitation could have significant implication for real-time applications.

2. The paper's claims are supported by strong empirical results. The experiments demonstrate substantial efficiency improvements in wall-clock time compared to conventional sequential sampling, as presented in Table 3. Furthermore, the authors validate their claim of exact sampling by showing that the generated samples are statistically indistinguishable from the samples obtained with traditional methods, using metrics like MMD on several benchmark datasets.

3. The proposed sampling algorithm is principled and rests on a solid theoretical foundation. The core of the method is a general technique for computing the necessary rejection constant by constructing piecewise linear upper and lower bounds for the target and proposal densities. The authors also introduce theoretical results on finding rejection constant efficiently.

4. A significant strength of this work is its generality. The proposed algorithm is model-agnostic and can be used as a drop-in addition to many existing parametric TPP models. The paper's theoretical framework is designed to handle a wide variety of distributions.

**Weaknesses:**

1. A significant limitation of this approach is its reliance on a closed-form, parametric density function. Computing the rejection constant, particularly the general method in Section 3.2, requires the ability to evaluate both the target density and its derivative to find inflection points and construct the necessary linear bounds. This constraint makes the approach unapplicable to many intensity-based or diffusion-based TPP models.
2. The paper's claim in Section 3.4 that the hidden states $h_{I+j}$ for the proposed events can be processed "in parallel" is questionable and highly model-dependent. If a model generates $h_{I+j}$ autoregressively, they cannot be processed in parallel.
3. It seems that the efficiency is also data-dependent. As the author acknowledges, the Taxi dataset, due to its dataset-specific properties, leads to a large rejection rate and inferior efficiency improvement relative to other datasets.

**Questions:**

Please see the weakness section.

---

> ### Author Response · Authors · 2025-11-14
>
> Thank you for your thoughtful review and constructive feedback. We appreciate the opportunity to clarify and strengthen our work. Below, we address each point in detail:
>
>
> **1. Reliance on a closed-form, parametric density function**
>
> Parametric densities are the standard in TPP literature and widely used in practice because they enable efficient training and inference. Our method builds on this common setting, making it broadly applicable to most existing models.
>
>
> **2. Parallel processing of hidden states**
>
> We agree that this point deserves clarification. While autoregressive models like RNNs compute hidden states sequentially by definition, modern implementations leverage fused kernels and optimized batching. This allows processing an entire sequence in a single uninterrupted pass, which is significantly faster than alternating between sampling and state updates step-by-step. Our approach exploits this efficiency: by proposing multiple events upfront, we can update all hidden states in parallel within one forward pass, achieving near-linear speedup compared to single-step updates.
>
>
> **3. Data-dependent efficiency (Taxi dataset)**
>
> Including the Taxi dataset was intentional to illustrate limitations of the base approach. Its deterministic mark transitions (alternating between two classes) result in zero acceptance beyond the first speculative step, which explains the lower efficiency gains. This behavior highlights that speculative sampling is most beneficial when marks exhibit uncertainty.
>
> We explored a simple extension: using a mixture of two previous mark distributions instead of one, which significantly improves performance on Taxi. Future work could adapt mixture weights dynamically or incorporate longer history. We omitted these refinements to keep the exposition focused on the core method, but we will mention this in the discussion section to acknowledge potential improvements.
>
>
> We hope this response helps convey that our method introduces a principled, model-agnostic approach for efficient TPP sampling, supported by theoretical guarantees and validated on diverse datasets. We believe the clarifications and additional context will strengthen the paper’s presentation and impact.

---

### Official Review · Reviewer_m8zr · 2025-10-31

**Soundness:** 3
**Presentation:** 2
**Contribution:** 3
**Rating:** 6
**Confidence:** 4

**Summary:**

This paper addresses the inefficiency of sequential sampling in autoregressive Temporal Point Processes (TPPs)—a critical limitation for real-time applications like finance and social network event modeling—by proposing a novel speculative sampling algorithm based on rejection sampling. TPPs model irregular event sequences (e.g., earthquakes, limit order book messages) but require sequential generation of events, which bottlenecks large-scale applications. The proposed method enables parallel sampling of multiple future events from existing parametric TPP models (no retraining/architectural changes) while guaranteeing exact sampling.

Key Contents.

1. The algorithm uses a pre-trained TPP encoder’s next-event distribution as a proposal to generate multiple future events in parallel. It accepts events until the first divergence between proposal and target distributions (derived by reprocessing proposed events with the encoder), ensuring exact sampling.

2. A universal method to compute rejection constants (critical for rejection sampling) via piecewise linear bounds on CDFs/PDFs.

3. Rigorous proofs confirm exact sampling (via valid rejection sampling) and bounded total variation error (≤δ) for categorical distributions with truncated constants, ensuring statistical correctness.

4. Experiments on 7 real datasets (e.g., Amazon, Earthquake, limit order books) show significant speedups (e.g., ~3.2x for Retweet data) with negligible quality loss (MMD/KL-divergence close to baseline). A financial application (limit order books) achieves average speculative steps of 2.82, validating practical utility.

5. Works with diverse encoders (GRU, Transformer, CNN) and decoders (log-normal mixture, exponential), making it compatible with most existing parametric TPPs.

**Strengths:**

The paper is easy to follow, and its main idea is clear and straightforward. The paper demonstrates notable originality by addressing a core inefficiency in TPP sampling—sequential generation bottlenecks—through a novel speculative sampling framework that avoids modifying existing models. Unlike prior works (e.g., Gloeckle et al., 2024; Zeng et al., 2023) that require retraining TPPs to predict multiple steps, this method leverages pre-trained encoders’ proposal distributions to generate parallel future events, with acceptance based on target-proposal divergence . A key innovation is the universal rejection constant calculation: using piecewise linear bounds on densities (exploiting convex/concave regions) to handle common TPP distributions (exponential, Gamma) and mixtures—an approach distinct from envelope methods (Gilks & Wild, 1992) that only apply to log-concave cases . This fills a gap for non-retrainable, high-frequency TPP applications (e.g., limit order books) where efficiency and model compatibility are critical.

In addition, the experimental results are substantial and adequate.

**Weaknesses:**

The main weakness of this paper is its presentation. Some parts are not clear enough. See my comments in "Questions" section.

**Questions:**

1. In Section 3.4, the renewal sampling could be made more clearly. For example, the authors can add more descriptions or formula to explain the generation from $(\tau_{i+j}, x_{i+j})$ to $(\tau_{i+j+1}, x_{i+j+1})$.

2. In "sample acceptance" of Section 3.4, the author should make it mathematically clear. What is the formula of the target distribution $p^*$ and the formula of the proposal $p$.

3. In Table 1, what does it mean by "commonly used distribution" in TPP models? Does it refer to the distribution for modeling the gap time $ \tau_{i+1} - \tau_i$?

4. In the experimental section, what does "sampling quality" mean? Could author give out the exact mathematical definition of the "quality"?

5. Could the author give some intuitive explanations for why top-k does not affect the sampling quality?

6. In the introduction of TPP, it is more formal to define $\mathcal H_i$ by the filtration of all past events.

**Details Of Ethics Concerns:**

No Concern.

---

> ### Author Response · Authors · 2025-11-14
>
> Thank you for your positive review and constructive feedback. Below, we address each point in detail:
>
>
> **1. Clarifying renewal sampling in Section 3.4**
>
> Thank you for highlighting this. We agree that additional clarity will benefit readers. In our method, there is no direct deterministic transition from $(\tau\_{i+j}, x\_{i+j})$ to $(\tau\_{i+j+1}, x\_{i+j+1})$. Instead, both are sampled in parallel from a shared proposal distribution (lines 238-242). Subsequently, the encoder processes these proposed events to compute the target distribution (lines 243-247). The implicit interaction between consecutive events occurs here: the encoder considers the entire sequence and applies a causal function to produce outputs. Each event is then independently flagged for acceptance, and the procedure terminates at the first rejection (lines 248-259). We will revise Section 3.4 to include a more explicit description.
>
>
> **2. Mathematical clarity for “sample acceptance”**
>
> We appreciate this suggestion. The formulas for both the proposal distribution $p$ and the target distribution $p^\star$ are provided in Table 1. The standard implementation assumes conditional independence between inter-event times and marks: $p(\tau, x) = p(\tau)p(x)$,  where $p(\tau)$ follows the distribution in Table 1 and $p(x)$ is categorical. Our approach is model-agnostic and supports alternative parameterizations. The exact distribution parameters are generated by a neural network, and the computation of $p$ and $p^\star$ is detailed in Section 3.4. We will make this connection more explicit in the revised manuscript.
>
>
> **3. Meaning of “commonly used distribution” in Table 1**
>
> Yes, this refers to distributions commonly employed to model the gap time $\tau\_{i+1}-\tau\_i$ in TPP literature. These distributions are the predominant choice in TPP literature.
>
>
> **4. Definition of “sampling quality”**
>
> Thank you for raising this point. There is no universal definition of sampling quality; however, one widely used metric is Maximum Mean Discrepancy (MMD), which we define in Appendix D.5. MMD provides a rigorous measure of similarity between distributions, where lower values indicate higher quality. Besides MMD, there are other metrics such as KL and LLR, both of which we use in our paper. What we refer to “sampling quality” is a combination of standard metrics in TPP literature (Table 2), comparing samples generated by traditional sequential sampling and our speculative method. These metrics collectively assess whether both sampling procedures produce statistically equivalent distributions. We will make this definition clearer in the revision.
>
>
> **5. Intuition for why top-k does not affect sampling quality**
>
> If the process is renewal, any choice of top-k yields an equivalent sampling process for both marks and inter-event times.
>
> Beyond this, for inter-event times, consecutive distributions typically exhibit only minor shifts, so permitting a rejected sample has negligible impact on the overall distribution. In contrast, a strict top-1 strategy can be overly restrictive, often rejecting samples due to discrepancies in the tails between the proposal and target distributions (see also discussion in Section 3.1).
>
> Figure 3 reinforces this intuition by showing that even for Hawkes processes (non-neural model), speculative sampling across multiple future steps preserves the underlying distributional properties. We will expand this discussion in the revision to make these points clearer.
>
>
> **6. Formal definition of $\mathcal{H}\_i$**
>
> We agree that defining $\mathcal{H}\_i$ via filtration is more formal. Our choice followed conventions in neural TPP literature, prioritizing simplicity.
>
>
> We appreciate your insightful feedback and will incorporate these clarifications to strengthen the paper’s presentation. We believe the clarifications will make the paper even stronger without altering its core contributions.

---

### Official Review · Reviewer_4Lnf · 2025-11-02

**Soundness:** 2
**Presentation:** 2
**Contribution:** 2
**Rating:** 4
**Confidence:** 2

**Summary:**

This paper proposes a speculative sampling scheme for temporal point processes (TPPs), inspired by speculative decoding methods used in large language models. The method aims to accelerate sampling from autoregressive TPP models by generating multiple future events in parallel using rejection sampling, without retraining or architectural modifications. The authors derive rejection constants for common distributions, provide theoretical proofs for bounding the acceptance rate, and report empirical speedups on several datasets.

**Strengths:**

1. The paper identifies a relevant problem — the inefficiency of sequential sampling in autoregressive TPPs — and attempts to address it using ideas from speculative decoding.

2. The proposed algorithm is conceptually simple and can be implemented without retraining existing models.

3. Theoretical analysis (Section 3) is relatively clear, providing proofs for the bounding procedure.

**Weaknesses:**

1. Limited Novelty and Conceptual Contribution:

1)The proposed approach is a straightforward adaptation of speculative decoding to temporal point processes. There is no fundamental theoretical or algorithmic innovation beyond applying rejection sampling to a different data modality.

2) The main “novelty” (deriving rejection constants for specific distributions) is technical but not conceptual — such derivations follow standard bounding techniques from rejection sampling literature.

2. Theoretical Claims Are Overstated:

1) The claim of “exact sampling” is questionable: the acceptance rule relies on upper/lower bounds that are themselves approximations (piecewise linear). The proofs (Appendix A) ensure boundedness, not exact equivalence to the target process.

2) The treatment of mixture distributions is hand-wavy: the proof assumes shared grid convexity and linearity, which can break under multimodal densities (e.g., log-normal mixtures).

3) The relationship between the proposed “speculative” method and standard multi-step Monte Carlo sampling is not rigorously differentiated.

**Questions:**

1. Is the rejection constant estimation stable across distributions with heavy tails (e.g., log-normal with large σ)?

2. Could this approach be extended to continuous-time diffusion TPPs

---

> ### Author Response · Authors · 2025-11-14
>
> Thank you for your review and valuable feedback.
>
> ## Novelty
>
> Unlike prior approaches, such as speculative sampling in LLMs or traditional Ogata thinning, our method introduces a fundamentally different paradigm. These existing techniques, while innovative and impactful, operate under design principles tailored to their respective problems. For example, LLMs fundamentally rely on two separate models working together, that’s their core contribution. Our approach doesn’t use anything like that.
>
> In contrast, we propose a model-agnostic, theoretically grounded adaptation of speculative decoding for temporal point processes (TPPs). Our framework is equally novel yet fundamentally different, offering rigor and generality that sets it apart.
>
> ## Theoretical claims
>
> >The acceptance rule relies on upper/lower bounds that are themselves approximations
>
> The bounds we use are not approximations; they are strict upper and lower bounds that guarantee a valid rejection constant. This ensures we never underestimate $M$, and therefore, Theorem 3.1 does not involve approximate sampling.
>
> What you may be referring to is the deliberate introduction of a controllable error for efficiency. For example, in the case of categorical distributions (Section 3.1, lines 137–151), exact sampling is possible but impractical because low-probability classes can make $M$ arbitrarily large. We show that bounding the total variation distance by $< \delta$ allows better estimates of $M$ without compromising performance.
>
> Importantly, this approximation is optional. Exact sampling is always possible, but introducing a small, controlled approximation improves runtime without hurting empirical results. This is an inherent property of sampling and is common in practice (e.g., LLMs and MDMs also use approximate sampling).
>
> >The treatment of mixture distributions is hand-wavy: the proof assumes shared grid convexity and linearity, which can break under multimodal densities
>
> Our treatment of mixture distributions is rigorous and provides a strict bound on $M$.
>
> If two functions have bounding functions, the sum of the original functions is bounded by the sum of their bounding functions. We use piecewise linear functions to bound densities exactly.
>
> For mixtures, each component has its own grid points. It is straightforward to take the union of these grids and define piecewise linear bounds on the shared grid. Both components remain correctly bounded, and their sum is trivial to compute.
>
> We formalize this in Lemma A.2 (any density can be exactly bounded by piecewise linear functions) and Lemma A.3 (combining mixture components). Figure 2 illustrates this approach and confirms that multimodality does not pose an issue.
>
> >The relationship between the proposed “speculative” method and standard multi-step Monte Carlo sampling is not rigorously differentiated
>
> Standard Monte Carlo sampling typically uses Ogata's thinning algorithm (Section 2.2). Thinning relies on an upper bound on the intensity function, which is often loose, and samples points sequentially, accepting or rejecting based on the true intensity versus the bound. This process is inherently slow.
>
> In contrast, our method samples multiple future points in parallel from the proposal distribution. We then compute target distributions for these points and determine rejection constants for each step in parallel. Each step is flagged as accepted or rejected, and we accept until the first rejection.
>
> The key differences are:
>
> *   We do not use rejection sampling to generate samples directly; instead, we use it to compare distributions and accept multiple steps.
> *   We compare distributions via upper and lower bounds, unlike Monte Carlo, which compares a single point's intensity to an upper bound.
> *   Our tighter bounds lead to better acceptance rates and improved efficiency.
>
> ## Questions
>
> 1.  Is the rejection constant estimation stable across distributions with heavy tails?
>
> Yes, it is stable. In fact, log-normal with large $\sigma$ is easier to handle because its curve is gentler and can be bounded more effectively. Small $\sigma$ values produce sharp peaks, which can lead to higher rejection rates for small offsets between consecutive distributions, regardless of bound quality.
>
> 2. Could this approach be extended to continuous-time diffusion TPPs?
>
> Diffusion models lack parametric densities, so our method cannot be applied directly. One possible extension would involve matching full diffusion paths, as they define tractable denoising distributions, but this would not yield exact or efficient sampling.
>
> We appreciate your detailed feedback and will incorporate clarifications to strengthen the presentation. We hope this response helps convey that our method offers a principled, parallel sampling approach for TPPs, providing theoretical guarantees and substantial practical speedups-while addressing a key bottleneck in real-world applications.

---

### Meta-Review · Area_Chair_JNQm · 2025-12-29

**Summary:**

1. **Novelty and Conceptual Contribution**:

   * Reviewers questioned the novelty of the proposed approach, suggesting it was primarily an adaptation of speculative decoding from large language models to temporal point processes (TPPs) without a fundamentally new theoretical or algorithmic contribution. The novelty was mostly seen in the technical aspects, like deriving rejection constants for specific distributions. However, the fundamental methodology was perceived as similar to existing methods in rejection sampling.

2. **Theoretical Claims and Proofs**:

   * Several reviewers raised concerns about the theoretical guarantees, particularly regarding the claim of "exact sampling." The rejection constant estimation relies on approximations and piecewise linear bounds, which some felt could lead to inaccuracies in certain cases, especially with multimodal distributions. Some reviewers also questioned the treatment of mixture distributions and its general applicability across different TPP models.

3. **Algorithm’s Applicability and Limitations**:

   * The approach's reliance on parametric density functions limits its applicability, as many intensity-based or diffusion-based TPP models may not fit this requirement. Furthermore, the need for a closed-form parametric density function was viewed as a significant limitation in practice.
   * The parallel processing claim was also challenged, particularly in models that generate events autoregressively. Some reviewers felt that the parallel processing step might be infeasible or inefficient depending on the model's structure.

4. **Empirical Results and Data-Specific Concerns**:

   * While the paper demonstrated significant empirical speedups, the efficiency improvements were data-dependent. The Taxi dataset, for instance, showed inferior performance due to its deterministic nature, highlighting that the speculative sampling method is most beneficial when there is significant uncertainty in the data. This was seen as a limitation in the generalizability of the method.

5. **Clarity and Presentation**:

   * Some reviewers found the presentation unclear in certain sections, such as in the explanation of renewal sampling and the sample acceptance process. They requested additional descriptions and formulae to clarify these aspects, which could enhance the overall readability and transparency of the paper.

6. **Future Work and Extensions**:

   * Several reviewers suggested potential future work, including extending the method to more complex or high-dimensional distributions, improving the computational efficiency of rejection constant computation, and exploring other methods like backtracking or partial resampling for certain datasets.

**Reviewer Concerns:**

### Unaddressed Concerns:

1. **Novelty and Conceptual Contribution**: The reviewers remained concerned that the approach was primarily an adaptation of speculative decoding methods for TPPs rather than a fundamentally new innovation. The rebuttal argued that the framework is novel and distinct, but this concern persists, particularly regarding the perceived lack of conceptual novelty.

2. **Theoretical Claims and Approximations**: The authors claimed strict upper and lower bounds in their method, but several reviewers continued to question whether these approximations—especially in multi-modal and heavy-tailed distributions—could lead to inaccurate results or violate the exact sampling guarantee. The rebuttal addressed some of these points but did not fully resolve the theoretical concerns.

3. **Algorithm’s Applicability to Non-Parametric TPP Models**: The method's reliance on parametric density functions remains a limitation that was not fully alleviated by the rebuttal. While the authors clarified that their method is suited to many existing models, the issue of its limited applicability to diffusion-based or intensity-based TPP models continues to be a concern.

4. **Data Dependency and Performance with Certain Datasets**: The efficiency of the method was shown to vary significantly across datasets, such as with the Taxi dataset. While the authors acknowledged the issue and proposed future improvements, the reviewers' concerns about the method's dependence on data characteristics remain valid.

### Addressed Concerns:

1. **Clarity of Theoretical Foundation and Mathematical Formulations**: The authors provided additional clarification regarding the derivation of rejection constants, the renewal sampling process, and sample acceptance. These clarifications have likely resolved some of the concerns about the paper's mathematical clarity, particularly regarding Section 3.4.

2. **Parallel Processing of Hidden States**: The authors clarified how modern implementations of autoregressive models can still achieve parallel processing, even if the models are inherently sequential. This point was explained more clearly in the rebuttal, addressing the concern raised by the reviewers.

**Reviewer Scores:**

### Reviewer 4Lnf:

* **Original Score**: 4 (Marginally below the acceptance threshold)
* **Revised Score**: Likely to remain the same or slightly lower. Despite the rebuttal addressing some technical details, the concerns about novelty, theoretical claims, and the method's limited applicability to non-parametric models would still weigh heavily. The reviewer would likely maintain their marginally negative assessment due to these unresolved issues.

### Reviewer m8zr:

* **Original Score**: 6 (Marginally above the acceptance threshold)
* **Revised Score**: Likely to stay the same. While the rebuttal addressed the theoretical concerns and some clarity issues, the reviewer’s concerns about the data-dependent performance and the limited novelty of the method would likely cause them to keep their score around the threshold, recognizing the solid empirical results but maintaining reservations about its broader impact.

### Reviewer v7DL:

* **Original Score**: 6 (Marginally above the acceptance threshold)
* **Revised Score**: Likely to remain the same or slightly higher. This reviewer appreciated the empirical results and theoretical foundation, but the concerns regarding data dependence and applicability to non-parametric models would still influence their score. However, they might slightly increase their score if they were convinced by the rebuttal’s clarifications on some theoretical aspects, but they would still have lingering doubts.

### Reviewer koMV:

* **Original Score**: 4 (Marginally below the acceptance threshold)
* **Revised Score**: Likely to stay the same or slightly lower. Despite the rebuttal addressing some clarity issues, the concerns about piecewise-linear approximation and the reliance on parametric densities would likely lead this reviewer to maintain their score below the acceptance threshold. They might even lower their score if they felt the rebuttal didn’t fully resolve the concerns about approximation accuracy and generalizability to other models.

---

### Decision · Program_Chairs · 2026-01-26

Reject